# On the evolution of sub- and super-saturated water uptake of secondary organic aerosol in chamber experiments from mixed precursors

Yu Wang[1,*,#], Aristeidis Voliotis[1], Dawei Hu[1], Yunqi Shao[1], Mao Du[1], Ying Chen[2], Judith Kleinheins[3], Claudia Marcolli[3], M. Rami Alfarra[1,4,5], Gordon McFiggans[1,*]

[1]Centre for Atmospheric Science, Department of Earth and Environmental Sciences, The University of Manchester, Manchester M13 9PL, UK

[2]Exeter Climate Systems, University of Exeter, Exeter, EX4 4QE, UK

[3]Institute for Atmospheric and Climate Science, ETH Zurich, 8092, Zurich, Switzerland

[4]National Centre for Atmospheric Science, Department of Earth and Environmental Sciences, The University of Manchester, Manchester, M13 9PL, UK

[5]Environment & Sustainability Center, Qatar Environment & Energy Research Institute, Doha, Qatar

*Correspondence to*: Yu Wang (yu.wang@manchester.ac.uk); Gordon McFiggans (g.mcfiggans@manchester.ac.uk)

[#]Now at Institute for Atmospheric and Climate Science, ETH Zurich, 8092, Zurich, Switzerland

## Abstract.

To better understand the chemical controls of sub- and super-saturated aerosol water uptake, we designed and conducted a series of chamber experiments to investigate the evolution of SOA particle physicochemical properties during photo-oxidation of single and mixed biogenic (α-pinene, isoprene) and anthropogenic (*o*-cresol) volatile organic compounds (VOCs) in the presence of ammonium sulphate seeds. During the six-hour experiments, the cloud condensation nuclei (CCN) activity at super-saturation of water (0.1 ~ 0.5 %), hygroscopic growth factor at 90% RH, and non-refractory $PM_1$ chemical composition were recorded concurrently. Attempts to use the hygroscopicity parameter κ to reconcile water uptake ability below and above water saturation from various VOC precursor systems were made, aiming to predict the CCN activity from the sub-saturated hygroscopicity. The thermodynamic model AIOMFAC was used to simulate κ values of model compound mixtures to compare with the observation and to isolate the controlling factors of water uptake at different RH.

The sub- and super-saturated water uptake (in terms of both $\kappa_{HTDMA}$ and $\kappa_{CCN}$) were mainly controlled by the SOA mass fraction which depended on the SOA production rate of the precursors, and the SOA composition played a second-order role. For the reconciliation of $\kappa_{HTDMA}$ and $\kappa_{CCN}$, the $\kappa_{HTDMA}$ / $\kappa_{CCN}$ ratio increased with the SOA mass fraction and this was observed in all investigated single and mixed VOC systems, independent of initial VOC concentrations and sources. For all VOC systems, the mean $\kappa_{HTDMA}$ of aerosol particles was ~ 25 % lower than the $\kappa_{CCN}$ at the beginning of the experiments with inorganic seeds. With the increase of condensed SOA on inorganic seed particles throughout the experiments, the discrepancy of $\kappa_{HTDMA}$ and $\kappa_{CCN}$ became weaker (down to ~ 0 %) and finally the mean $\kappa_{HTDMA}$ was ~ 60 % higher than $\kappa_{CCN}$ on average when the SOA mass fraction approached ~ 0.8. As indicated by AIOMFAC model simulations, non-ideality alone cannot fully explain the κ discrepancy at high SOA mass fraction (0.8). A good agreement in $\kappa_{CCN}$ between model and observation was achieved by doubling the molecular weight of the model compounds or by reducing the dry particle size in the CCN counter. This indicates that the evaporation of semi-volatile organics in the CCN counter together with non-ideality could have led to the observed κ discrepancy. As a result, the predicted CCN number concentrations from the $\kappa_{HTDMA}$ and particle number size distribution were ~ 10 % lower than CCN

counter measurement on average at the beginning, and further even turned to an overestimation of ~ 20 % on average when the SOA mass fraction was ~ 0.8. This chemical composition-dependent performances of the κ-Köhler approach on CCN prediction can introduce a variable uncertainty in predicting cloud droplet numbers from the sub-saturated water uptake, the influence of which on models still needs to be investigated.

# 1 Introduction

Aerosol-cloud interactions, that is how aerosol particles influence cloud formation, largely influence Earth radiation budget and the current climate projections (Boucher et al., 2013; Lohmann and Feichter, 2005; Bellouin et al., 2020). Thus, an accurate prediction of cloud condensation nuclei (CCN) number from aerosol properties is essential for investigating aerosol-cloud interactions in climate models. However, the reliability of cloud condensation nuclei (CCN) activity predicted from the aerosol hygroscopic growth under sub-saturated condition remains unresolved, e.g. (Cruz and Pandis, 1998; Vanreken et al., 2005; Huff Hartz et al., 2005; Prenni et al., 2007; Petters et al., 2009; Wex et al., 2009; Ervens et al., 2007; Good et al., 2010b; Liu et al., 2018). One of the main knowledge gaps is the precise determination of CCN activity involving complex organic aerosols. A large portion of organic aerosols are secondary organic aerosol (SOA) (Zhang et al., 2007; Jimenez et al., 2009), formed from oxidation of gaseous volatile organic compounds (VOCs) via gas-particle partitioning (Hallquist et al., 2009) and aqueous-phase reactions (Ervens et al., 2011; Kuang et al., 2020b). Although the organic aerosol components are less soluble and consequently less hygroscopic than the referenced inorganic compounds (e.g. sulphate, nitrate) (Alfarra et al., 2013; Mcfiggans et al., 2006; Kreidenweis and Asa-Awuku, 2014; Huff Hartz et al., 2005; King et al., 2009), they can play an important role in the cloud formation globally (Liu and Wang, 2010; Rastak et al., 2017) due to its ubiquitous large fraction (20 - 90 %) in fine particulate matter mass (Kanakidou et al., 2005; Jimenez et al., 2009; Zhang et al., 2007). Nevertheless, our understanding of its hygroscopicity and CCN activity remains uncertain, due to the wide range of solubility, volatility and complex composition of organic compounds from different sources (Hallquist et al., 2009; Goldstein and Galbally, 2007; Shrivastava et al., 2017).

Previous laboratory reconciliation studies of aerosol hygroscopicity and CCN activity were mainly focused on experiments investigating the nucleation of SOA from single biogenic VOC oxidation e.g. (Prenni et al., 2007; Wex et al., 2009; Petters et al., 2009; Alfarra et al., 2013; Liu et al., 2018; Zhao et al., 2016; Duplissy et al., 2008), from anthropogenic VOC (Zhao et al., 2016; Liu et al., 2018; Prenni et al., 2007) and a few from biogenic-anthropogenic VOC mixtures (e.g. Zhao et al., 2016). However, the findings are not consistent. For the biogenic SOA, most studies found that the single hygroscopicity parameter ($\rho_{ion}$, κ) from CCN activity were 20 - 70 % higher than the that from sub-saturated hygroscopicity, using oxidation of representative biogenic precursors, such as monoterpenes (Wex et al., 2009; Liu et al., 2018; Zhao et al., 2016; Prenni et al., 2007) and sesquiterpenes (Huff Hartz et al., 2005). They speculated that the higher measured CCN activity of the biogenic SOA may be caused by the complex composition and variable properties, such as the suppressed surface tension below that of the pure water induced by organic surfactants (Wex et al., 2009), the presence of sparingly soluble organic compounds (Petters et al., 2009; Prenni et al., 2007), non-ideality-driven liquid-liquid phase separation (Liu et al., 2018) or joint influences of these factors. In contrast, Duplissy et al. (2008) found a good reconciliation of hygroscopicity parameter κ between the hygroscopicity at 95 % RH and CCN activity of the SOA from α-pinene oxidation. Moreover, Good et al. (2010b) found that the agreement of κ reconciliation of the SOA from α-pinene ozonolysis was influenced by the use of three different custom-built Hygroscopicity Tandem Differential Mobility Analyser (HTDMA) for sub-saturated hygroscopicity measurements and the absence/presence of inorganic seed. For the anthropogenic or biogenic-anthropogenic mixed SOA, Zhao et al. (2016) observed a smaller discrepancy of κ than for the biogenic SOA, but the measured CCN activity was still higher than the sub-saturated hygroscopicity (> 20 %). In contrast, Liu et al. (2018) found no discrepancy for anthropogenic SOA.

Clearly the complexity of aerosol chemical composition can propagate to their water uptake behaviour. Consequently, our understanding of the chemical controls on the sub- and super-saturated water uptake is still limited, especially for the evolution of the multi-component organic-inorganic systems. To further improve our understanding on chemical controls of water uptake of multi-component aerosol particles, we designed and performed a series of chamber experiments to investigate the evolution of the chemical composition, the sub- and super-saturated water uptake of SOA from single and mixed VOCs in the

presence of ammonium sulphate seed. The novelty of the project is its design to investigate SOA formation from single to mixed precursors whereas previous studies mainly focused a single precursor (Voliotis et al., 2022). The interaction of the mixed precursor could influence SOA properties, therefore this study takes a further step of lab studies towards the real atmosphere where thousands of precursors are existing and reacting at the same time even the chemical regime and complexity of the chamber studies could deviate from the real atmosphere. The ultimate goal of this paper was to explore the change and controlling factors in the water uptake of multicomponent seeded particles as they transformed through the oxidation of the various mixed VOC systems.

# 2 Materials and method

## 2.1 Experiment design

A series of chamber experiments were designed and conducted at Manchester Aerosol Chamber (MAC) to investigate the impacts of mixing VOCs on the SOA formation mechanisms and aerosol physicochemical properties (e.g. chemical composition, volatility, water uptake). An overview of the overall project can be found in Voliotis et al. (2022). Briefly, this work builds on the concept explored in Mcfiggans et al. (2019) using a mixture of the biogenic SOA precursors, α-pinene and isoprene, extended to a ternary system by including *o*-cresol as an anthropogenic VOC. *o*-cresol is both directly emitted anthropogenically or naturally and is a first generation oxidation product of toluene, both being abundant aromatic VOCs observed in anthropogenic polluted areas (Seinfeld and Pandis, 2016). *o*-cresol is sufficiently close in reactivity towards OH radical with α-pinene and isoprene as to contribute comparable amounts of oxidation products to the mixture (Coeur-Tourneur et al., 2006; Iupac). Additionally, it is a moderate SOA yield compound (Henry et al., 2008), so any interactions in the mixture with the oxidation products of the other VOCs may lead to contrasting interactions to those in the binary high-yield α-pinene mixture with low-yield isoprene (Mcfiggans et al., 2019; Voliotis et al., 2022). VOCs were injected into the chamber with modest VOC/NO$_x$ ratio ranging 4 - 10 and the mixing ratio of VOCs were chosen such that they would have the same reactivity towards ·OH at the beginning of the experiment (though clearly

not necessarily after the commencement of photochemistry). In addition, ammonium sulphate particles were injected as seeds for SOA condensation considering its abundance in atmosphere (Seinfeld and Pandis, 2016). Characteristic experiments of each system are chosen and details of the initial conditions are shown in Table 1. Single VOC isoprene experiments were carried out, but not included in this study

since they had undetectable levels of SOA mass above our background under the neutrally-seeded conditions of our experiments with no noticeable change to hygroscopicity.

A detailed description and characterisation of the MAC facility (e.g. controlling condition stability, gas/particle wall loss, auxiliary mechanism, aerosol formation capability) can be found in Shao et al. (2022). Briefly, MAC consists of an 18 $m^3$ FEP Teflon bag supported by movable aluminium frames and

135 runs as a batch reactor. The chamber is mounted inside the enclosure where the air conditioning system can well control the temperature (25 ± 2 °C) and relative humidity (RH, 50 ± 5 %) in chamber in this study. For photochemistry experiments, two 6kW Xenon arc lamps (XBO 6000 W/HSLA OFR, Osram) and 5 rows (#16 for each row) of halogen bulbs (Solux 50W/4700K, Solux MR16, USA) are used to mimic the solar spectrum of mid-day of clear sky conditions in June in Manchester and the total actinic

flux between 290 and 600 nm was ~1/3 of the clear-sky solar radiation in Manchester (Shao et al., 2022). A reproducible cleaning protocol was conducted, including daily cleaning (fill chamber with ~1ppm of $O_3$ and stay overnight to remove reactive organics and perform automatic fill/flush physical cleaning cycles before and after experiments) and regular harsh cleaning with high concentration of $O_3$ under strong ultraviolet light. During an experiment, seed particles are injected and well mixed with a high flow

rate blower and kept well-mixed within chamber by the continual external agitation of conditioned air through the gap between the enclosure and chamber. Liquid VOCs (α-pinene, isoprene, $o$-cresol; Sigma Aldrich, GC grade ≥ 99.99 % purity) are injected with syringes through a heated glass bulb t in which the liquids can be vaporized immediately under ~ 80 °C and then flushed into chamber with high purity nitrogen (ECD grade, 99.997 %). $NO_x$ (as mostly $NO_2$ in this study) injection is controlled by a mass flow

controller and the desired RH in chamber is moderated by the mixing of water vapour and dry purified clean air to chamber to adjust the desired RH condition. A series of instruments were deployed to record gas precursors (VOC, $NO_x$, $O_3$) and physicochemical properties of seeded SOA. Details of key instruments used in this study can be found in Sec. 2.2.

Table 1. Experimental initial conditions of the various single and mixed biogenic and anthropogenic VOC systems
photochemistry in the presence of ammonium sulphate seed.

| Date | VOC type | [VOC]$_0$ (ppbV) | VOC/NO$_x$ | Seed conc. (ug/m$^3$)[a] |
|---|---|---|---|---|
| 2019.03.29 | α-pinene | 309 | 7.2 | 67.6 |
| 2019.04.17 | α-pinene | 155 | 4.4 | 46.2 |
| 2019.07.13 | α-pinene | 103 | 5.7 | 55.4 |
| 2019.04.12 | o-cresol | 400 | n.a. | 40.9 |
| 2019.04.19 | o-cresol | 200 | 5.0 | 56.0 |
| 2019.07.10 | o-cresol | 133 | 4.9 | 38.1 |
| 2019.04.08 | α-pinene/isoprene | 237 (155/82) | 9.9 | 50.5 |
| 2019.04.23 | α-pinene/o-cresol | 355 (155/200) | 5.9 | 42.5 |
| 2019.04.24 | o-cresol/isoprene | 282(82/200) | n.a. | 57.0 |
| 2019.07.30 | α-pinene/isoprene/o-cresol | 191 (103/55/133) | 3.7 | 45.9 |

[a] calculated mass concentration from volume concentration from DMPS with a density of 1.77 g cm$^{-3}$.

n.a. means no available data due to instrument failure.

## 2.2 Measurements

The measured aerosol particles are dried with a Nafion® drier (Perma Pure, MD-110-12, Toms River, NJ, USA) to RH < 30 % before introduced to the following instruments. The sub-saturated water uptake of aerosol particles was measured by a custom-built Hygroscopicity Tandem Differential Mobility Analyser (HTDMA) (Good et al., 2010a). The HTDMA is used to determine aerosol growth factor (GF) at a certain RH. Principally, sampled aerosol particles are dried and then selected by the first Differential Mobility Diameter (DMA1) to get monodisperse aerosol particles at given size (D$_0$), which further are humidified at 90 % RH in this study. The humidified aerosol particles enter the second DMA and a Condensation Particle Counter (CPC) in order to determine the size distributions. The HTDMA was calibrated and its performance was validated by (NH$_4$)$_2$SO$_4$ before and after the campaign following the method of Good et al. (2010a). Finally, the growth factor probability density function and mean growth

factor (GF) were retrieved using TDMA$_{inv}$ method developed by Gysel et al. (2009). To track particle growth, the measured particle size increased from 75 nm up to 300 nm, depending on the geometric mean diameter of aerosol populations during SOA formation evolution processes in various VOC systems.

The super-saturated water uptake of aerosol particles, that is the ability to activate to CCN, was measured by a DMT continuous flow CCN counter (Roberts and Nenes, 2005). In this study, the CCN counter was coupled with a DMA and a CPC to obtain the fraction of size-resolved aerosol particles activating to CCN ($F_A$) at a certain supersaturation. Briefly, the DMA was used to select monodisperse dried aerosol particles (RH < 30 %), which are fed into the CCN counter and CPC in parallel to count the activated and total number concentrations of aerosol particles, respectively. During the experiments, DMA scans from 20 to 550 nm with 20 size bins, splitting the flow to direct the size-selected aerosol particles through the CCN counter and a CPC to measure the CCN and total particle number concentrations, respectively. The supersaturation ratio of the CCN counter is usually set to 0.5 % at the beginning of experiments. With ongoing SOA formation, the aerosol particles grow. To derive a reliable activation curve with enough particle number concentration around the activation size, the set supersaturation ratio decreases accordingly down to 0.1 % during experiments, depending on how fast the SOA forms. The time resolution for each measurement is 10 min. $F_A$ as a function of the dry particle size ($D_0$) was derived from the ratio of the activated and total aerosol particles concentrations with a correction of DMA multiple charge. Finally, the particle size at 50 % activation (Dc$_{CCN}$) was identified through a sigmoid fit of $F_A$-$D_0$ curve, which was assumed to be the critical diameter at the critical supersaturation (Sc$_{CCN}$). CCN counter was calibrated and its performance was validated by $(NH_4)_2SO_4$ before and after the campaign following the procedure in Good et al. (2010a).

The chemical composition of the non-refractory PM$_1$ components (NR-PM$_1$, including ammonium NH$_4$, sulphate SO$_4$, nitrate NO$_3$, SOA) was measured by a High-Resolution Time-of-Flight Aerosol Mass Spectrometer (HR-ToF-AMS, Aerodyne Research Inc., USA). Detailed instrument descriptions can be found elsewhere (Decarlo et al., 2006; Jayne et al., 2000; Allan et al., 2004; Allan et al., 2003). During the experiment period, HR-ToF-AMS was calibrated and its performance was validated following the standard procedures (Jayne et al., 2000; Jimenez et al., 2003). In addition, to obtain the size-resolved

chemical composition, a polystyrene latex sphere (PSL) calibration was performed to obtain the relationship between vacuum aerodynamic particle size and its velocity following the protocol provided at http://cires1.colorado.edu/jimenez-group/wiki/index.php/Field_Data_Analysis_Guide (last access: 24-01-2022).

For the conversion of AMS vacuum aerodynamic diameter to mobility diameter, firstly, we estimated the density of the non-refractory aerosol particles using simple mixing rule shown in equation [1] assuming the density of ammonium sulphate (1.77 g/cm$^3$) and SOA (1.4 g/cm$^3$).

$$\rho_{est} = \rho_{AS}(1 - F_{m,SOA}) + \rho_{SOA}F_{m,SOA} \quad [1]$$

$F_{m,SOA}$ is the mass fraction of the SOA. Then, this estimated density is used to calculate the mobility diameter as shown in equation [2] (Zhang et al., 2005).

$$D_m \approx \frac{D_{va}}{\rho_{est}} \quad [2]$$

For the MR$_{SOA/PM}$ uncertainty, the choice of SOA density can introduce uncertainty to $\rho_{est}$, with implications for the mobility diameter. Previous studies found that the SOA density can range from 1.2 to 1.65 g/cm$^3$ (Kostenidou et al., 2007; Alfarra et al., 2006; Varutbangkul et al., 2006; Nakao et al., 2013). For example, Kostenidou et al. (2007) reported that the estimated density of SOA from $\alpha$-pinene, $\beta$-pinene, d-limonene are 1.4-1.65 g/cm$^3$. Nakao et al. (2013) investigated the SOA from 22 different precursors with a wide range of carbon number (C5-C15) and found their density ranging from 1.22 to 1.43 g/cm$^3$, negatively related to their molecular size. In this study, considering the three precursors we used, we take a medium value of density (1.4 g/cm$^3$). To calculate the uncertainty of the SOA density on MR$_{SOA/PM}$, we recalculated with the minimum (maximum) density, 1.2 (1.65) g/cm$^3$, the MR$_{SOA/PM}$ changes within $\pm$ 10%.

## 2.3 κ-Köhler approach

A single parameter κ is used to bridge the sub- and super-saturated water uptake, which is readily applied to predict cloud properties from aerosol physicochemical properties in climate models (Fanourgakis et al., 2019). However, it should be noted that the non-ideality of solution (e.g. the sparingly soluble SOA, molecular and ionic interactions), the potential influence of SOA on surface tension and the difference in co-condensation of condensable vapours through the systems will influence the results as previously

discussed (Wex et al., 2009; Prenni et al., 2007; Hu et al., 2018), and which will be further discussed in Secs. 3.4 and 3.5.

The hygroscopicity parameter κ from sub-saturated HTDMA and super-saturated CCN counter are referred as $\kappa_{HTDMA}$ and $\kappa_{CCN}$, respectively. $\kappa_{HTDMA}$ was calculated directly through Eq. [1-2] with the measured GF and dry particle size $D_0$.

$$S(D) = a_w exp\left(\frac{4\sigma M_w}{RT\rho_w D}\right) \quad [3]$$

$$\kappa = \frac{V_w}{V_s}\left(\frac{1}{a_w} - 1\right) = \frac{D^3 - D_0^3}{D_0^3}\left(\frac{1}{a_w} - 1\right) \quad [4]$$

$$D = D_0 GF \quad [5]$$

For CCN measurement, $\kappa_{CCN}$ was derived from the computed κ-Dc-Sc relationship at surface tension of water and temperature of 298.15 K in Petters and Kreidenweis (2007). Here, Dc and Sc represent the dry

diameter of aerosol particle and the critical supersaturation ratio of water vapour (maxima of the Köhler curve) to activate it to CCN.

Where S(D) is the supersaturation ratio or RH at sub-saturated condition. D and $D_0$ represents the dry and wet particle diameter, respectively. $a_w$, σ, $M_w$, $\rho_w$ are activity, droplet surface tension, molecular weight, and density of water, respectively. R and T represents the universal gas constant and absolute temperature,

respectively. GF is the growth factor at 90 % RH measured by HTDMA.

## 2.4 κ-modeling with AIOMFAC

To study the influence of non-ideality on $\kappa_{HTDMA}$ and $\kappa_{CCN}$, model calculations were performed using the group contribution model AIOMFAC (Zuend et al., 2008; Zuend et al., 2010; Zuend et al., 2011; Zuend and Seinfeld, 2012; Zuend and Seinfeld, 2013) to calculate activity coefficients. Since the real SOA composition is complex and the exact chemical composition is unknown, the goal here was not to simulate the composition as realistically as possible but to create mixtures of model compounds that cover the experimental range of hygroscopicity. The hygroscopicity depends solely on the hydrophilicity of the substance (affecting the activity coefficients) and the number of solute molecules in a particle (affecting the mole fraction) which is determined by their molecular weight. Reactivity is not considered in thermodynamic modelling. The hydrophilicity of a substance depends on its chemical composition, most importantly on the number of polar functional groups while the exact arrangement of the functional groups is of minor relevance. Thus the hydrophilicity can be captured by the O:C ratio, which also determines the tendency for liquid-liquid phase separation in aerosol particles (Song et al., 2012). Therefore, by examining model compound mixtures covering broadly the range of experimentally determined O:C ratios and realistic molecular weights, the possible range of κ values can be investigated without the necessity to replicate the real mix of chemical structures.

The mixtures chosen here contained between two and eight different organic compounds, most of them α-pinene oxidation products, mainly with carboxyl (-COOH), hydroxyl (-OH) and/or keto (C=O) functionalities. The average O:C ratio of the organic mixtures ranges between 0.36 and 0.95, while the O:C ratio of the experimental SOA ranges between $0.36 \pm 0.03$ and $0.69 \pm 0.05$ (Wang et al., 2021b). The average molar mass of the mixtures was varied in a broad range of 173 – 478 g/mol. High molar masses were achieved by artificially dimerizing the original model compounds by doubling each subgroup of the molecule, similar to the approach by Zuend and Seinfeld (2012). To isolate the effect of non-ideality from co-condensation effects, all substances were assumed to be non-volatile and gas-particle partitioning was not explicitly modelled. Therefore, the selected substances were chosen to have sufficiently large molecular weights for allowing partitioning to the condensed phase. The lower bound of the average molar masses is reached by model compound mixtures that match the experimentally measured volatility

distribution of the SOA (Voliotis et al., 2021). Even lower average molar masses or O:C ratios would not alter the drawn conclusion as can be seen in Section 3.5 and Figure S3 of the supporting information (SI). Table S1 and S2 in the SI list the monomeric model compounds and all mixture compositions.

For each mixture, the water-partitioning and potential liquid-liquid phase separation was calculated with AIOMFAC using the algorithm of Zuend and Seinfeld (2013). In this algorithm, the calculations are performed for a bulk system. To obtain the corresponding relative humidity in equilibrium with the droplet (S), the water activity was multiplied with the Kelvin effect based on the wet diameter $D$ at this water activity, following Köhler-theory (Köhler, 1936) as shown in Eq. [3]. $\kappa_{HTDMA}$ and $\kappa_{CCN}$ were calculated according to Eq. [4] (Petters and Kreidenweis, 2007). $V_w$ and $a_w$ are taken from the AIOMFAC output at $S(D) = 90\%$ and $Sc$, respectively.

# 3 Results and Discussion

## 3.1 Bulk and size-dependent chemical composition

Figures 1 and 2 show the bulk NR-PM$_1$ species and size-resolved organic mass fraction (MR$_{SOA/PM}$) measured by HR-ToF-AMS, respectively. At the beginning of experiments before illumination (-1 - 0 h), seed particles are mainly comprised of sulphate with a small contribution from nitrate (Max. 5 % - 16 % of NR-PM$_1$) in all investigated VOC systems. The observed nitrate was mainly inorganic ammonium nitrate and the organic nitrate was statistically insignificant (a detailed estimation method and discussion can be found in Wang et al. (2021a)). Considering the small fraction of nitrate in the inorganic seed particles in this study and comparable water uptake ability with sulphate (Kreidenweis and Asa-Awuku, 2014), it may be expected that the overall hygroscopicity and CCN activity will be highly related to the MR$_{SOA/PM}$. After initiating illumination, the condensable organic vapours were formed from VOCs photo-oxidation, which further condensed on the inorganic seed particles yielding SOA. Therefore, an increasing MR$_{SOA/PM}$ over time was observed, as shown in Fig. 1. As different VOC systems have different SOA yield and reactivity with oxidants (Voliotis et al., 2022), the mass and the production rate of SOA varied

with the VOC systems. After a six-hour photochemistry for the single VOC systems, the $MR_{SOA/PM}$ approached $0.88 \pm 0.01$, $0.82 \pm 0.01$, $0.62 \pm 0.01$, $0.71 \pm 0.01$, $0.56 \pm 0.02$ and $0.52 \pm 0.02$ (last 0.5 h of experiments, avg. ± std.) in the α-pinene, 50 % reactivity α-pinene, 33 % reactivity α-pinene, $o$-cresol, 50 % reactivity $o$-cresol and 33 % reactivity $o$-cresol systems, respectively. For the binary and ternary systems, the $MR_{SOA/PM}$ was $0.79 \pm 0.01$, $0.82 \pm 0.01$, $0.32 \pm 0.01$ and $0.78 \pm 0.01$ in the α-pinene/isoprene, α-pinene/$o$-cresol, $o$-cresol/isoprene, and α-pinene/$o$-cresol/isoprene, respectively. Moreover, a size-dependent chemical composition was observed, with a higher $MR_{SOA/PM}$ for particles at 75/100 nm than the 200/300 nm particles in all investigated VOC systems (as shown in Fig. 2). This indicates that the chemical composition is not uniform across the size distribution. As the inorganic compounds are much more hygroscopic than the SOA (Kreidenweis and Asa-Awuku, 2014; Prenni et al., 2007; Alfarra et al., 2013; Alfarra et al., 2012), aerosol hygroscopicity and CCN activity will vary with $MR_{SOA/PM}$. Considering measured dry size differences between the HTDMA and CCN counter, size-resolved chemical composition has been used to ensure that the paired $\kappa_{HTDMA}$ and $\kappa_{CCN}$ for measurement reconciliation are with comparable $MR_{SOA/PM}$.

## 3.2 Aerosol hygroscopicity under sub-saturated conditions

The GF at 90 % RH was measured by a HTDMA and hygroscopicity parameter ($\kappa_{HTDMA}$) was calculated using the κ-Köhler approach (Petters and Kreidenweis, 2007) for all the investigated VOC systems as shown in Fig. 3. Before the photochemistry with inorganic seed only, the GF at 90 % RH ($\kappa_{HTDMA}$) for the 75 / 100 nm aerosol particles were 1.65 - 1.72 (0.45 - 0.50) in all VOC systems. This result is comparable with the predicted GF ($\kappa_{HTDMA}$) of 1.71 (0.51) of the $(NH_4)_2SO_4$ using AIOMFAC with the assumption of non-ideality. After the commencement of photochemistry, the $MR_{SOA/PM}$ increased over time. Consequently, the GF ($\kappa_{HTDMA}$) decreased accordingly due to the less hygroscopic nature of SOA compared with the one of inorganic compounds (Kreidenweis and Asa-Awuku, 2014; Prenni et al., 2007; Alfarra et al., 2013; Alfarra et al., 2012; Varutbangkul et al., 2006).

As expected, the rate of change and magnitude of the GF ($\kappa_{HTDMA}$) decreases over time depends on the change of $MR_{SOA/PM}$ in all VOC systems. For example, for the α-pinene system, the $MR_{SOA/PM}$ increased substantially from ~ 0 to 0.72 within an hour of the experiment (as shown in Fig. 1a), correspondingly, the GF ($\kappa_{HTDMA}$) decreased from 1.65 - 1.72 (0.45 - 0.50) to ~ 1.15 (~ 0.1) (as shown in Fig. 3a). In comparison, for the o-cresol/isoprene system, it took six hours for the $MR_{SOA/PM}$ to increase to 0.33, and

accordingly, the GF ($\kappa_{HTDMA}$) decreased slowly to 1.44 - 1.53 (0.28 - 0.36) after the six-hour experiment. Moreover, consistent with the observed higher $MR_{SOA/PM}$ for smaller size in Sec. 3.1, Fig. 3 shows evidence that the GF ($\kappa_{HTDMA}$) is size-dependent, with up to ~ 0.2 (~ 0.1) lower in 100 nm than in 200 nm aerosol particles, measured adjacently. This is consistent with the non-uniform size-dependent particle chemical composition in our chamber studies. Consideration of size-resolved chemical composition is

very important for the aerosol physical and optical properties where both chemical composition and particle size can play a role.

## 3.3 CCN potential under super-saturated conditions

CCN activity above water saturation was simultaneously recorded by CCN counter during the

335 experiments of all investigated VOC systems. Fig. 4 shows the relationship of the critical supersaturation of water vapour (Sc), the dry particle size and the $\kappa_{CCN}$. It provides the required Sc to activate 50 % of a given size of dry particles (Dc$_{CCN}$), for which this CCN activation potential can be represented by a single hygroscopicity parameter ($\kappa_{CCN}$) (Petters and Kreidenweis, 2007). At the beginning of experiments before photochemistry, the $\kappa_{CCN}$ was mainly 0.55 - 0.65 in all investigated VOC systems, which is comparable

with predicted κ of 0.61 from AIOMFAC. After initiating photochemistry, a declining trend of $\kappa_{CCN}$ over time was observed as the continuous condensation of less hygroscopic / CCN-active SOA, consistent with the trends of sub-saturated water uptake in Sec. 3.2. For example, for the α-pinene system as shown in Fig. 4a, the $\kappa_{CCN}$ decreased from 0.64 to ~ 0.1 within an hour whereas the $\kappa_{CCN}$ decreased from 0.55 to 0.23 after the six-hour oxidation for the o-cresol/isoprene system. This significant differences between

different VOC systems are highly related to the production rate of SOA and the corresponding change of $MR_{SOA/PM}$ over time. It is worth noting that the set-point Sc in CCN counter was changed from 0.1 - 0.5 %

during the experiments to follow the particle growth and ensure sufficient data points are collected for the activation curve to accurately determine the $Dc_{CCN}$.

## 3.4 CCN prediction from the sub-saturated conditions

This section illustrates the reconciliation of the aerosol hygroscopicity and CCN activity, and its relationship with the aerosol chemical composition in various VOC systems to investigate the performance of the κ-Köhler approach in predicting CCN activity from sub-saturated aerosol hygroscopicity. As shown in Sec. 3.1-3.2, the aerosol chemical composition is size-dependent. It is essential to ensure the chemical composition is comparable for HTDMA and CCN measurements for the reconciliation study if their measured dry particle sizes are different. Therefore, we selected the synchronized HTDMA/CCN data pairs only when the 10-min moving average of $MR_{SOA/PM}$ for the measured particle sizes were within 5 %. An example of selected data pairs in the α-pinene/isoprene/*o*-cresol system is shown in Fig. S1 of SI. In addition to the hygroscopicity parameter κ, the critical diameter ($Dc_{Hpre}$) was predicted from $\kappa_{HTDMA}$ following the κ-$Dc$-$Sc$ relationship in Sec. 2.3 under the critical supersaturation of the paired CCN measurement. Further, by assuming all particles larger than $Dc_{Hpre}$ be activated at the given $Dc_{Hpre}$, the CCN number was predicted based on the $Dc_{Hpre}$ and particle number size distribution.

Fig. 5 shows a summary of (a) $\kappa_{HTDMA}$, (b) $\kappa_{CCN}$, (c) $\kappa_{HTDMA}/\kappa_{CCN}$, (d) $\kappa_{HTDMA}-\kappa_{CCN}$, (e) $Dc_{Hpre}/Dc_{CCN}$, and (f) $Nccn_{Hpre}/Nccn_{CCN}$ as a function of the organic mass fraction in various VOC systems (except for α-pinene and 33 % α-pinene systems due to CCN instrument failure). Similar trends of the investigated parameters as a function of $MR_{SOA/PM}$ were observed in all VOC systems. As shown in panel a-b, the hygroscopicity parameter $\kappa_{HTDMA}$ and $\kappa_{CCN}$ decreased with the increase of $MR_{SOA/PM}$ in all VOC systems, indicating aerosol particles became less hygroscopic and CCN-active modified by the increasingly condensed SOA. For a summary of all data points binned with a $MR_{SOA/PM}$ of 0.1, the black solid circles and grey lines represent the average and standard deviation of the categorized data points. The overall $\kappa_{HTDMA}$ ($\kappa_{CCN}$) declined from 0.46 ± 0.02 (0.61 ± 0.07) to 0.14 ± 0.03 (0.09 ± 0.01) when the $MR_{SOA/PM}$ increased from ~ 0 to ~ 0.8.

In addition to the overall trend, the $\kappa_{HTDMA}$ ($\kappa_{CCN}$) at the same $MR_{SOA/PM}$ were different in the different VOC systems which indicated that the SOA composition played a second-order role in the hygroscopicity (CCN activity). A higher $\kappa_{HTDMA}$ ($\kappa_{CCN}$) of the multi-component SOA-inorganic mixtures at the same $MR_{SOA/PM}$ indicated a higher $\kappa$ of the SOA, according to the Zdanovski-Stokes-Robinson (ZSR) mixing rule of $\kappa$ demonstrated in Petters and Kreidenweis (2007). In this study, the $\kappa_{HTDMA}$ ($\kappa_{CCN}$) (indicating a higher $\kappa$ of the SOA), in the α-pinene/isoprene/o-cresol and 33 % o-cresol systems were the highest, which are higher than other VOC systems by 0 - 0.2 (0 - 0.3), depending on the $MR_{SOA/PM}$. In contrast, the $\kappa_{HTDMA}$ ($\kappa_{CCN}$) in o-cresol and 50 % reactivity o-cresol were usually the lowest at the same level of $MR_{SOA/PM}$, whereas the 50 % reactivity α-pinene, α-pinene/isoprene and o-cresol/isoprene seated in the middle. Previous studies found the sub-saturated aerosol water uptake ($\kappa$) increases with chemical aging of SOA from single precursor oxidation and showed a positive relationship with SOA oxidation state (e.g. O:C ratio or f44, fraction of m/z 44 in total organic signal) (Jimenez et al., 2009; Massoli et al., 2010; Lambe et al., 2011; Zhao et al., 2016; Duplissy et al., 2011; Kuang et al., 2020a), but no clear relationship involving multiple precursors with various oxidation state (Alfarra et al., 2013; Zhao et al., 2016). In addition, Wang et al. (2019a) found that the positive relation between water uptake at supersaturated conditions and oxidation state (O:C) can be attributed to lower molecular weight of organic species rather than higher solubility at higher oxidation level. To illustrate the relationship between $\kappa$ of SOA and the oxidation state, the $\kappa_{org}$ was deduced with ZSR method and the $\kappa$ of ammonium sulphate from AIOMFAC assuming volume additivity. Two main messages are shown in Fig. S2. Firstly, the calculated $\kappa_{org}$ from HTDMA and CCN counter varied with VOC systems ranging from -0.2 to 0.2. The ZSR method assumes that components are independent and the water uptake by individual components are additive. Therefore, the negative values of the $\kappa_{org}$ indicates the existence of interactions between inorganic and organic substances and thus results in less water uptake than the case without interactions in ZSR method (Zardini et al., 2008). Secondly, the calculated $\kappa_{org}$ at sub- and supersaturated conditions showed no clear relationship with oxidation state of SOA (f44) when various VOC systems are compared, which is consistent with previous studies involving multiple precursors (Alfarra et al., 2013; Zhao et al., 2016). Other factors might have influenced the results and warrant further investigations, such as organic mass loading, molecular weight (Cappa et al., 2011; Petters et al., 2017), solubility (Petters et al., 2009; Ruehl

and Wilson, 2014; Huff Hartz et al., 2006), surface tension (Ovadnevaite et al., 2017; Bzdek et al., 2020; Ruehl et al., 2016; Lowe et al., 2019) and co-condensation (Kulmala et al., 1993; Topping et al., 2013; Hu et al., 2018), and will be discussed in Sec. 3.5.

Panel c-d in Fig. 5 shows the ratio ($\kappa_{HTDMA}/\kappa_{CCN}$) and the absolute difference ($\kappa_{HTDMA}-\kappa_{CCN}$) of $\kappa$ derived from HTDMA and CCN counter as a function of $MR_{SOA/PM}$. Interestingly, a clear co-increase of $\kappa_{HTDMA}/\kappa_{CCN}$ ($\kappa_{HTDMA}-\kappa_{CCN}$) with the $MR_{SOA/PM}$ was observed in all VOC systems. The overall $\kappa_{HTDMA}/\kappa_{CCN}$ for all VOC systems increased from $0.76 \pm 0.08$ to $1.62 \pm 0.26$ with the $MR_{SOA/PM}$ increasing from ~ 0 to ~ 0.8, and correspondingly, the $\kappa_{HTDMA}-\kappa_{CCN}$ increased from $-0.15 \pm 0.06$ to $0.05 \pm 0.02$. This

means the averaged $\kappa_{HTDMA}$ was ~ 25 % (16 % - 32%) lower than $\kappa_{CCN}$ with inorganic compounds at the beginning of the experiments, but this discrepancy decreased down to ~ 0 with the increasing $MR_{SOA/PM}$ and even became higher than $\kappa_{CCN}$ by ~ 60 % (36 % - 88 %) at $MR_{SOA/PM}$ of ~ 0.8 (as shown in Fig. 5c). These results indicated that the performances of $\kappa$-Köhler approach on the reconciliation study of sub- and super-saturated water uptake varied with the $MR_{SOA/PM}$.

The discrepancy in the $\kappa_{HTDMA}$ and $\kappa_{CCN}$ can influence the prediction of CCN activity from sub-saturated hygroscopicity ($\kappa_{HTDMA}$) using the $\kappa$-Köhler approach. As shown in Fig. 5 e, the predicted critical diameter ($Dc_{Hpre}$) was 5 - 20 % (avg. ~ 10 %) higher than the measured $Dc_{CCN}$ at $MR_{SOA/PM}$ of 0.02, and the $Dc_{Hpre}/Dc_{CCN}$ decreased gradually to 0.8 - 1 (avg. ~ 0.9) as $MR_{SOA/PM}$ approached 0.8. As a result, the predicted CCN number concentration from sub-saturated water uptake was underestimated by 0 - 20 %

(avg. ~ 10 %) at $MR_{SOA/PM}$ of 0.02. This underestimation of CCN number became weaker (averaged value almost down to ~ 0) with $MR_{SOA/PM}$ increased to 0.2 - 0.4 due to SOA condensation, and the underestimation even reversed to an overestimation by up to 40 % (avg. 20 %) with $MR_{SOA/PM}$ of ~ 0.8 (as shown in Fig. 5f). It is worth noting that the prediction of critical diameter and CCN number concentration from $\kappa_{HTDMA}$ are based on the concurrently measured critical supersaturation and particle

number size distribution. This dependence of $\kappa_{HTDMA}/\kappa_{CCN}$ ratio on chemical composition can have a varied impact on the uncertainty of the predicted CCN activity from sub-saturated $\kappa_{HTDMA}$ at different supersaturation ratio of water vapour and/or different particle number size distribution as measured above. Because the activated CCN number concentration is determined by all the three factors: the $\kappa_{HTDMA}$, water

supersaturation ratio and particle size distribution. If at different supersaturation ratio of water vapour
and/or different particle number size distribution as measured in this study, the uncertainty of the
predicted CCN activity from sub-saturated $\kappa_{HTDMA}$ can change. Indeed, this discrepancy trend between
$\kappa_{HTDMA}$ and $\kappa_{CCN}$ could introduce a varied impact on the CCN prediction, which needs further
investigations.

## 3.5 Analysis of the model results and discussion of the κ-discrepancy

As demonstrated above, the $\kappa_{HTDMA}$ was, on average, ~ 25 % lower than the $\kappa_{CCN}$ of the inorganic seeds
when the $MR_{SOA/PM}$ was ~ 0, which is consistent with the thermodynamic model results from AIOMFAC
with the assumption of non-ideality (both κ were 0.72 if assuming ideality).

To examine the influence of non-ideality at higher organic mass fractions, model calculations with
AIOMFAC were performed to explore whether the mean experimental $\kappa_{HTDMA}$ ($0.14 \pm 0.03$) and $\kappa_{CCN}$
($0.09 \pm 0.01$) at $MR_{SOA/PM} = 0.8$ can be reproduced by including non-ideality. To this purpose, 17 model
compound mixtures of average O:C ratios between 0.36 and 0.95 and average molar masses between 173
and 478 g/mol were designed, that cover the hygroscopicity range spanned by the SOA products. For
none of these mixtures the experimental $\kappa_{HTDMA}$ and $\kappa_{CCN}$ at $MR_{SOA/PM} = 0.8$ could be met. The trends of
the simulation results are exemplified in Fig. 6 for four out of the 17 mixtures, which combine low (O:C
= 0.36) and high (O:C = 0.66) oxidation with low and high molecular weights. Most of the low molecular
weight compounds are identified α-pinene oxidation products, while the high molecular weight
compounds are artificial dimers of the monomeric compounds. Further details regarding the four mixtures
can be found in the SI under mixture numbers 5 (red line in Fig. 6), 6 (yellow), 14 (blue) and 15 (cyan).
For the monomeric SOA with O:C = 0.66, a calculation assuming solution ideality (activity coefficients
set to one) was also performed. It can be seen that the assumption of solution ideality leads to an
overestimation of $\kappa_{HTDMA}$ and $\kappa_{CCN}$ for all organic mass fractions including the inorganic seed ($MR_{SOA/PM}$
of ~ 0). In AIOMFAC, the ideal aqueous ammonium sulphate solution is calculated as fully dissociated
into $2\,NH_4^+ + 1\,SO_4^{2-}$ (corresponding with van't Hoff factor of three) with activity coefficients set to one.

At activation, ideal solution conditions would be expected, as the particles are strongly diluted. However, for ammonium sulphate a large difference in $\kappa_{CCN}$ between the ideal and non-ideal model calculation can be observed. This difference suggests some association of the ions in solution, possibly to $N_2H_7^+$ and $HSO_4^-$ (Atwood et al., 2002). AIOMFAC accounts for concentration and composition dependent speciation of ammonium sulphate in solution through the activity coefficients, which have been adjusted

during the parameterization process to bring the model output in agreement with the experimental data (Zuend et al., 2008). Including non-ideality leads to an overall better agreement of $\kappa_{HTDMA}$ at all organic mass fractions (Fig. 6a). At high organic mass fractions ($MR_{SOA/PM} = 0.8$), best agreement of $\kappa_{HTDMA}$ is reached for the simulations with O:C = 0.66 irrespective of the molecular weight (i.e. monomers and dimers). In contrast to that, Fig. 6b shows the best agreement of $\kappa_{CCN}$ at $MR_{SOA/PM} = 0.8$ for the model

mixture with dimers with average O:C = 0.36, which is the one that agrees least with the observed $\kappa_{HTDMA}$ values. As a result, the $\kappa_{HTDMA}/\kappa_{CCN}$ (Fig. 6c) and $\kappa_{HTDMA}-\kappa_{CCN}$ (Fig. 6d) at $MR_{SOA/PM} = 0.8$ could not be reproduced with the model compound mixtures shown in this figure. Overall, only mixtures with dimers and low O:C ratios were able to match the experimental range of $\kappa_{CCN}$, yet, only dimer mixtures with rather high O:C ratios were able to fully match $\kappa_{HTDMA}$. Thus, among all 17 examined mixtures, none was

found where the modelled $\kappa_{HTDMA}$ and $\kappa_{CCN}$ values were both within the standard deviation range of the experimental values (see Fig. S3 in the SI), indicating that non-ideality alone cannot account for the discrepancy between $\kappa_{HTDMA}$ and $\kappa_{CCN}$.

Previous studies found that some organic compounds are strongly surface-active, and can lower the surface tension of the droplet below the value of pure water even at activation (Ovadnevaite et al., 2011;

Bzdek et al., 2020; Gérard et al., 2019). While the effect of a lowered surface tension on hygroscopic growth is negligible, assuming a lowered surface tension at supersaturated conditions would lead to a reduction in $Sc$. In the experiment, however, a higher $Sc$ was measured than $\kappa_{HTDMA}$ would suggest (see Fig. S4 in the SI). Therefore, a lowered surface tension cannot explain the observed discrepancy in $\kappa$ at high $MR_{SOA/PM}$. Calculating $\kappa_{CCN}$ with the assumption of a lower surface tension would even lead to a

higher $\kappa_{CCN}$ thus increasing the discrepancy rather than reducing it.

Thermodenuder measurements showed that the examined SOA contained a substantial fraction of semi-volatile compounds (Voliotis et al., 2021). Differences in the design of the HTDMA and CCN counter

could have influenced the fate of the semi-volatile compounds, thereby explaining the observed $\kappa$ discrepancy. The semi-volatile compounds in the gas phase (e.g. organics, $HNO_3$) can co-condense with water vapor on aerosol particles and enhance the water uptake (Rudolf et al., 1991; Rudolf et al., 2001; Hu et al., 2018; Topping et al., 2013; Wang et al., 2019b; Gunthe et al., 2021). This enhancement is more significant at higher relative humidity. In addition, aerosol particles grow larger at higher relative humidity in the CCN counter and dilute the solute concentrations in the particle phase, which further facilitates the partitioning of semi-volatile compounds into the particle phase, creating a positive feedback. Therefore, equal organics in the gas phase and equal temperature in both instruments would result in $\kappa_{CCN} > \kappa_{HTDMA}$, which contrasts with the observation in this study. However, if the gas phase is diluted or if the temperature is increased, semi-volatile compounds in the particle phase can also evaporate and thereby decrease the water uptake when re-equilibrating (Hu et al., 2018). The observed higher $\kappa_{HTDMA}$ than $\kappa_{CCN}$ can be explained, if the organic concentration in the gas phase was significantly higher in the HTDMA than in the CCN counter and/or if the temperature in the CCN counter was higher than in the HTDMA.

The sampled aerosols from the chamber were dried to RH < 30 % before splitting and entering the HTDMA and CCN counter. During the drying process, semi-volatile compounds can co-evaporate with water to the gas phase. Water vapour was then removed through the Nafion membrane, but this pre-treatment was the same for both instruments. In our setup, the sheath air flows of the two DMAs in the HTDMA are close-loop, which means that the sheath air is filtered and recirculated and will reach equilibrium with the sample air including gaseous organic compounds. In commercial CCN counters, the sheath air is produced by splitting the sample air and filtering it (Roberts and Nenes, 2005) and thus, contains organic gases. However, the DMA for size selection before the CCN counter uses dry clean air as sheath air, which can dilute the aerosol flow and thereby result in the evaporation of organic compounds. Gaseous organic substances can deposit on the filters in both instruments and deposited material from previous experiments can desorb or evaporate from the filters, which could have influenced the sheath air composition.

After selecting a given size of aerosol by the first DMA, the aerosol went through the conditioned humid environment. The temperature was decreased to 18 °C to reach the set RH in HTDMA (Good et al.,

2010a), which will facilitate the co-condensation of the semi-volatile compounds and the growth of the aerosol particles by lowering the saturation vapour pressure due to temperature drop (Hu et al., 2018). In contrast to that, the temperature in the CCN counter is designed to increase to keep a certain water saturation (Roberts and Nenes, 2005), which is not favourable for co-condensation and could even have led to evaporation of semi-volatile compounds. A loss of organic mass in the CCN counter is equivalent to a smaller dry diameter of the particles, which results in a higher critical supersaturation and thus a lower $\kappa_{CCN}$. A decrease of ~15% of the dry diameter could explain the observed $\kappa$ discrepancy at $MR_{SOA/PM} = 0.8$. Setting the volume loss equal to a mass loss, the 15 % decrease in diameter is equivalent to a 39% mass loss, which approximately corresponds to the total loss of the organic mass in the $log C^* = 2$ volatility bin and half of the mass in the $log C^* = 1$ volatility bin of the measured volatility distribution of $\alpha$-pinene SOA (Voliotis et al., 2021). Partial evaporation in the CCN counter is also in good agreement with the fact that only model mixtures with dimeric compounds were able to reproduce the observed $\kappa_{CCN}$. As the high average molar mass required to match $\kappa_{CCN}$ contradicts the measured volatility distribution, this gives further support to the assumption of a loss of molecules in the CCN counter, as this reduces the particle size and increases the average molar mass. Note that a higher molar mass of the organics has the same effect as the absolute loss of molecules, since both lead to an overall smaller number of organic molecules in the particle, which reduces the Raoult effect. Thus, solution non-ideality together with evaporation of semi-volatile compounds in the CCN counter is a plausible explanation of the observed discrepancy between $\kappa_{HTDMA}$ and $\kappa_{CCN}$. Further factors that may have biased the $\kappa$-measurements include co-condensation of semi-volatile compounds in the HTDMA, the dilution of the sheath air in the size-selection DMA before the CCN counter and the influence of the filtering on the sheath air composition in both instruments. This exemplifies how challenging the physicochemical characterization of semi-volatile organic aerosols is. Further investigations are needed to clearly quantify possible effects of co-condensation and evaporation of semi-volatile compounds in HTDMAs and CCN counters to support this explanation of observed $\kappa$ discrepancies.

## 4 Conclusions

In this study, we designed and performed a series of chamber experiments to improve our understanding of the chemical controls of the sub- and supersaturated water uptake in the evolution of the SOA formation from mixed precursors in the presence of ammonium sulphate seed. The yield and reactivity of the SOA precursors controlled the SOA production rate in different VOC systems, and therefore the increase of organic mass fraction ($MR_{SOA/PM}$). Our results showed that the $MR_{SOA/PM}$ is the main factor influencing the hygroscopicity and CCN activity in terms of $\kappa$, and the SOA composition plays a second-order role. At the same level of $MR_{SOA/PM}$, the order of overall $\kappa_{HTDMA}$ and $\kappa_{CCN}$, from highest to lowest,  were α-pinene/isoprene/o-cresol and 33 % o-cresol > α-pinene, α-pinene/isoprene and o-cresol/isoprene > o-cresol and 50 % reactivity o-cresol systems. There is no clear relationship between the $\kappa$ of SOA deduced by ZSR method and oxidation level (f44).

During the SOA formation process in all VOC systems, size-resolved chemical composition was observed, for which the smaller particles have higher $MR_{SOA/PM}$. To avoid the influences of composition differences on the reconciliation study of sub- and super-saturated water uptake, the synchronized HTDMA and CCN data pairs with a comparable chemical composition were selected according to the size-resolved chemical composition.

In the reconciliation, we found the discrepancy between $\kappa_{HTDMA}$ and $\kappa_{CCN}$ varied with the $MR_{SOA/PM}$. Consequently, the performance of the κ-Köhler approach on CCN activity prediction from sub-saturated condition also changed with the $MR_{SOA/PM}$. This trend was observed in all investigated VOC systems, regardless of the VOC sources and initial concentrations. For all investigated VOC systems, the averaged $\kappa_{HTDMA}/\kappa_{CCN}$ increased from $0.76 \pm 0.08$ to $1.62 \pm 0.26$ when the $MR_{SOA/PM}$ increased from ~ 0 to ~ 0.8, meanwhile the mean absolute difference ($\kappa_{HTDMA}$-$\kappa_{CCN}$) increased from $-0.15 \pm 0.06$ to $0.05 \pm 0.02$. To explain these trends, AIOMFAC model calculations for representative model mixtures were performed. The increasing $\kappa_{HTDMA}/\kappa_{CCN}$ with increasing $MR_{SOA/PM}$ cannot be explained by potential surface tension reduction of organics as this effect will yield higher $\kappa_{CCN}$ and even increase the discrepancy. The non-ideality of mixed organic-inorganic solutions and the different co-condensation or evaporation behaviour of semi-volatile organic substances in the two measurement setups could be plausible reasons for the

discrepancy. Further experimental investigations on how HTDMAs and CCN counters respond to condensable vapours are of great importance to better understand this discrepancy.

In addition, we estimated the influences of this $\kappa$ discrepancy trend on the prediction of CCN number concentration from the sub-saturated hygroscopicity ($\kappa_{HTDMA}$). The predicted mean CCN number concentration was underestimated by ~ 10 % at $MR_{SOA/PM}$ of ~ 0. This underestimation of CCN number disappeared with an increase of $MR_{SOA/PM}$ to 0.2 - 0.4 due to SOA condensation, and ultimately turned to an overestimation by ~ 20 % in average with $MR_{SOA/PM}$ of ~ 0.8. It is worth noting that the influences of the $\kappa$ discrepancy trend on CCN activity prediction were estimated based on the current measurements of critical supersaturation and particle number size distribution. Broader impacts of this chemical-dependent performance of the $\kappa$-Köhler approach in cloud properties prediction under various atmospheric conditions should be analysed in climate models to better project aerosol-induced climate effects.

## Data availability

The observational dataset of this study is available upon request from corresponding authors.

## Author contributions

Y.W. conceived this study. G.M., M.R.A., Y.W., A.V. and Y.S. co-designed the chamber experiments. Y.W., A.V., Y.S. and D.M. conducted the chamber experiments. D.H. offered in-kind trainings on operation and data analysis of HTDMA and CCN counter for Y.W. During chamber experiments, Y.W. performed HTDMA and CCN counter measurements used in this study, conducted data integration and analysis, and wrote the manuscript. Y.C. provided helpful discussions. J.K. and C.M. designed and analysed AIOMFAC model simulations. G.M. and M.R.A. proofread and improved the manuscript.

# Acknowledgement

Manchester Aerosol Chamber acknowledges the financial support from EUROCHAMP 2020. We acknowledge AMF/AMOF for providing SMPS instrument. Y.W. acknowledges the joint scholarship of The University of Manchester and Chinese Scholarship Council. M.R.A. acknowledges support by UK National Centre for Atmospheric Sciences (NACS) funding. A.V. acknowledges the Natural Environment Research Council (NERC) EAO Doctoral Training Partnership funding. J.K. acknowledges the Swiss National Foundation for funding (project number: 200021L_197149).

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

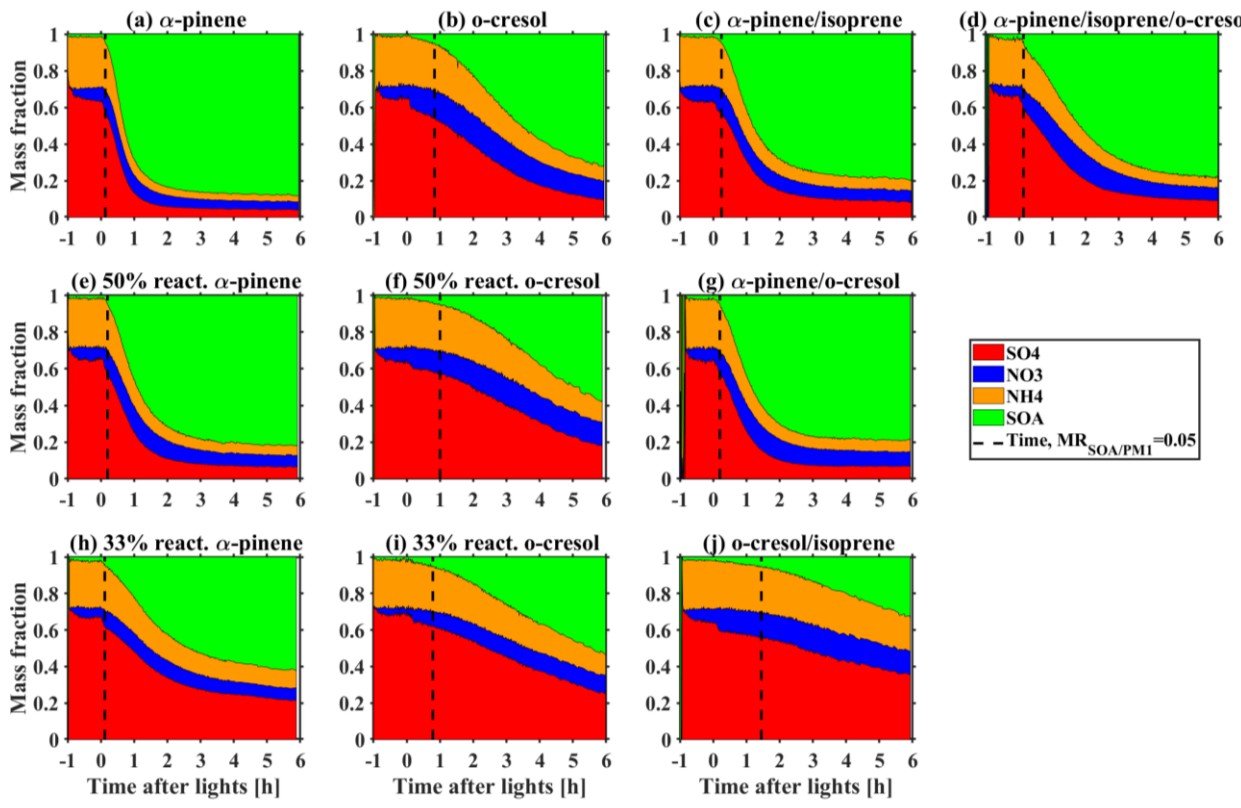

Figure 1. Mass fraction of chemical species in non-refractory PM$_1$ measured by HR-ToF-AMS during SOA formation evolution in various VOC systems.

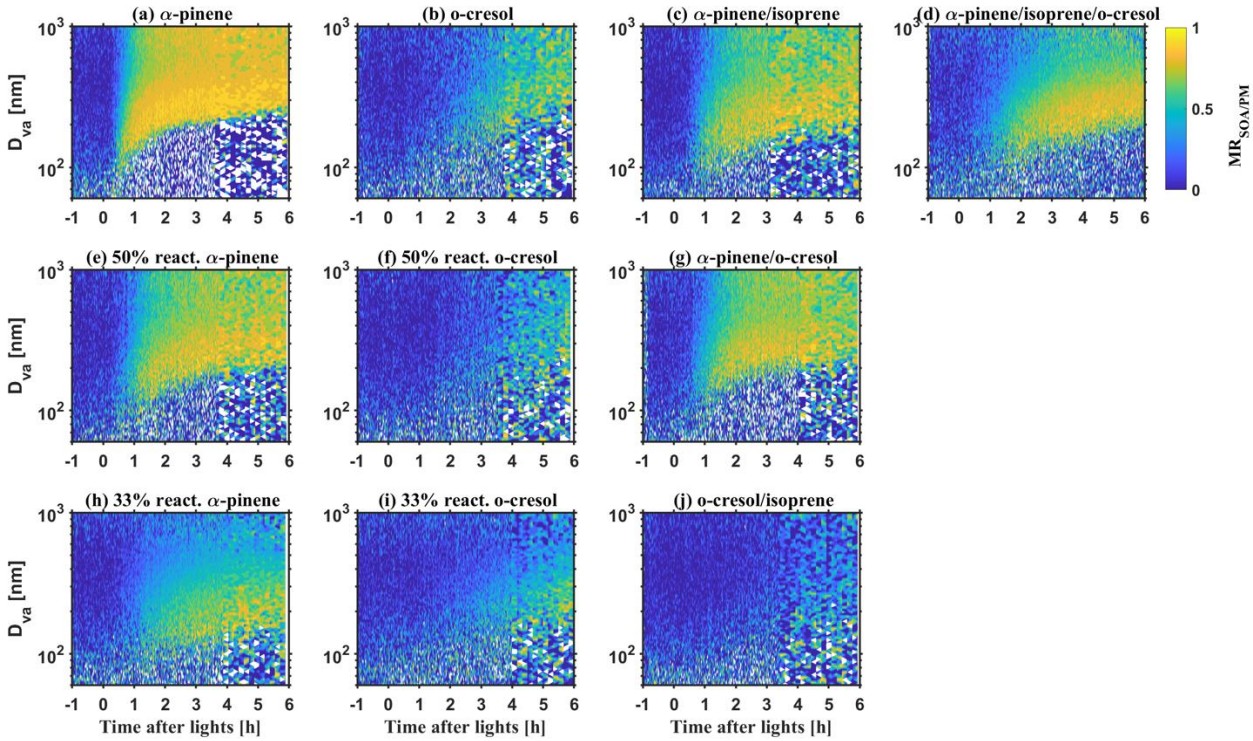

Figure 2. Size-resolved SOA mass fraction in non-refractory $PM_1$ ($MR_{SOA/PM1}$) measured by HR-ToF-AMS during SOA formation evolution in various VOC systems.

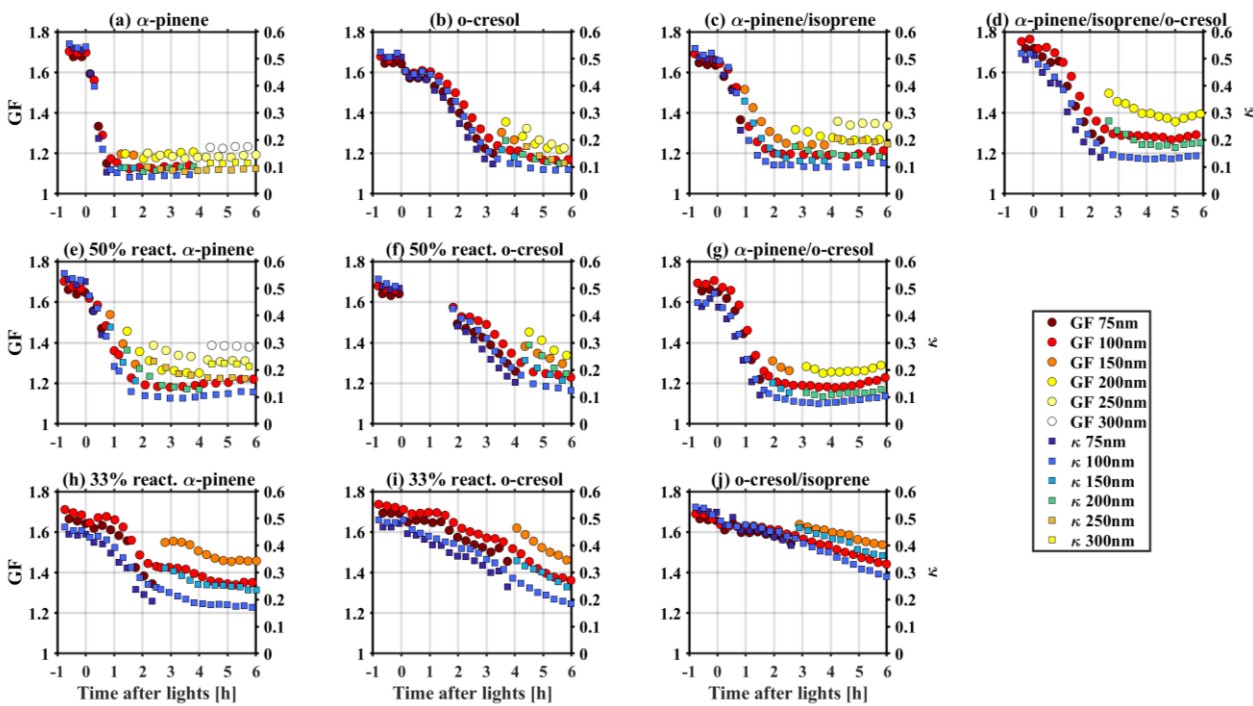

Figure 3. Time series of GF and κ at different measured particle size during SOA formation evolution in various VOC systems.

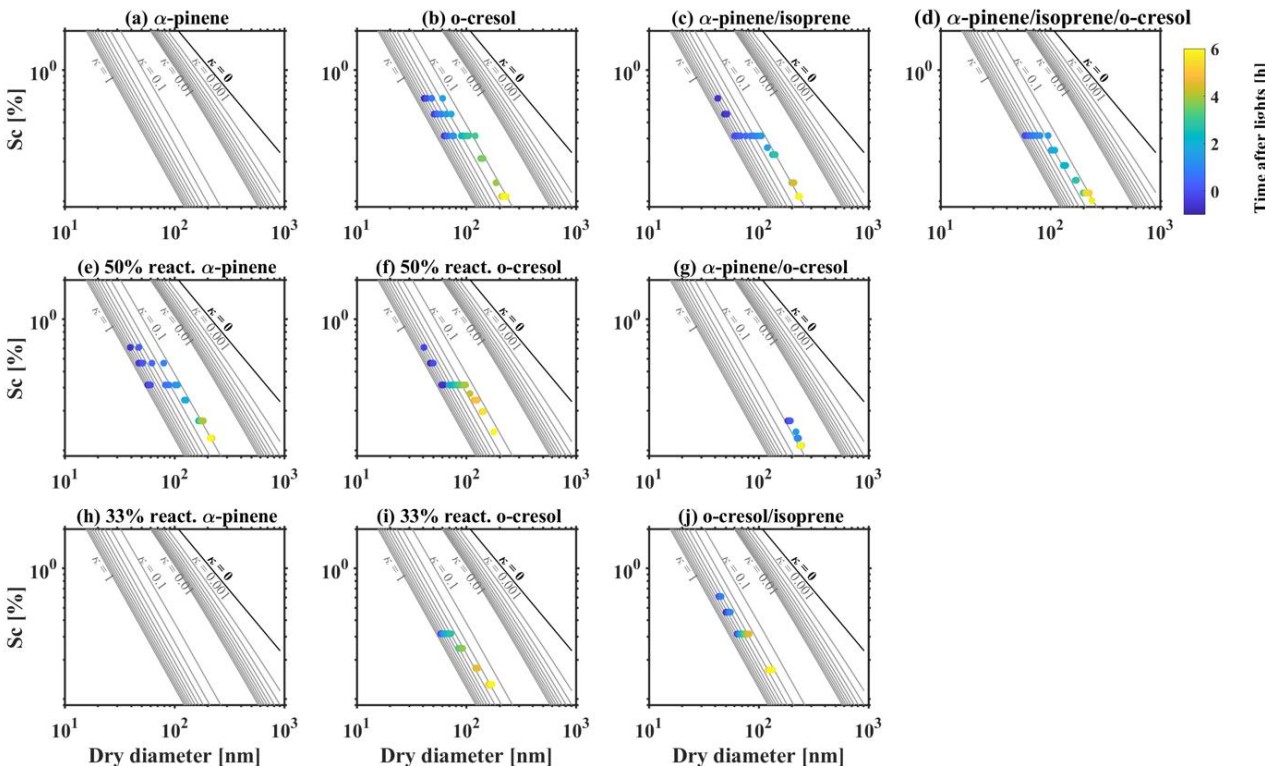

Figure 4. Critical supersaturation as a function of dry particle size ($D_{50}$) measured by CCN counter during SOA formation evolution in various VOC systems. Contour lines represent hygroscopicity κ, calculated by following the method in Petters and Kreidenweis (2007).

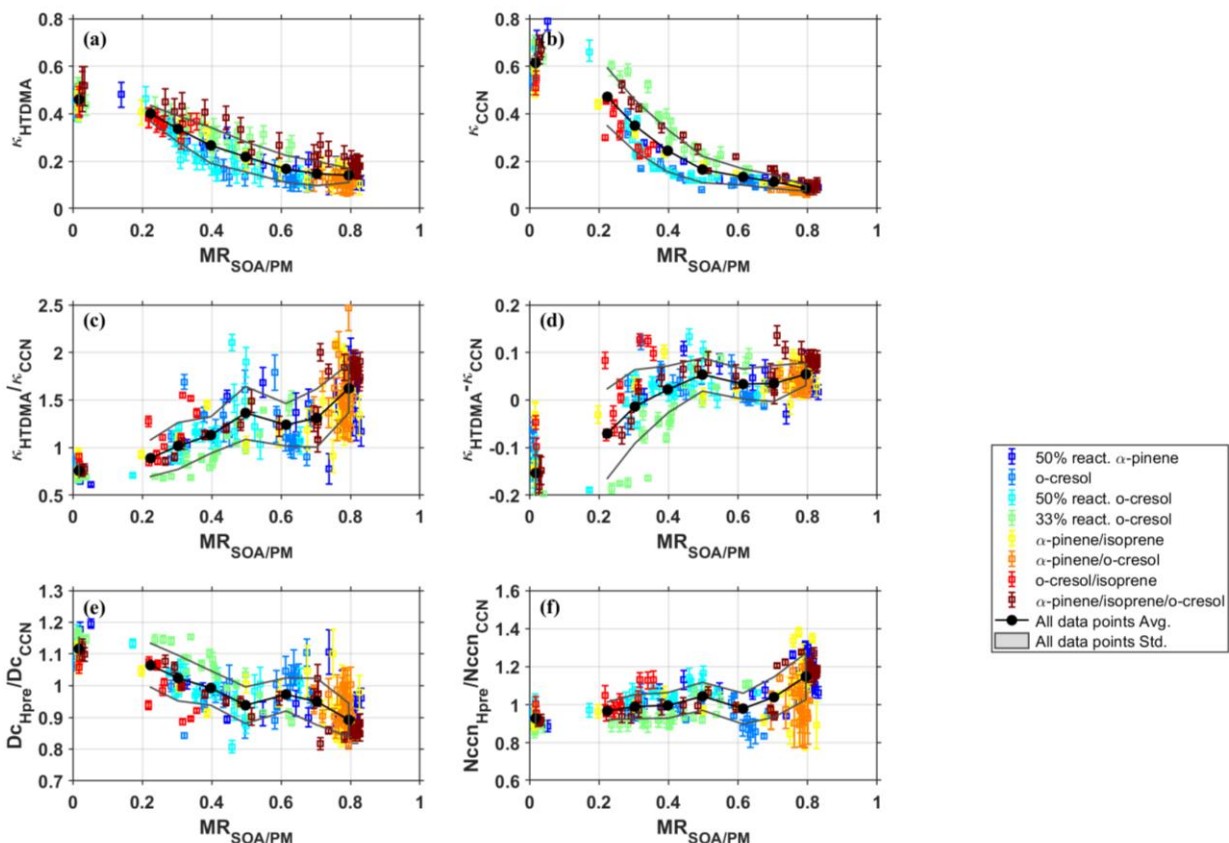

Figure 5. (a) $\kappa_{HTDMA}$, (b) $\kappa_{CCN}$, (c) $\kappa_{HTDMA}$ / $\kappa_{CCN}$, (d) $\kappa_{HTDMA}$ - $\kappa_{CCN}$, (e-f) critical diameter and CCN number concentration between HTDMA prediction using κ-Köhler theory and CCN measurement, as a function of $MR_{SOA/PM}$ in various investigated VOC systems. The errorbar of $\kappa_{HTDMA}$ and $\kappa_{CCN}$ in panel a and b represent measurement uncertainty following the method in Irwin et al. (2010). The uncertainty in $\kappa_{HTDMA}$ and $\kappa_{CCN}$ then propogate to the uncertainty of parameters shown in Panel c-f.

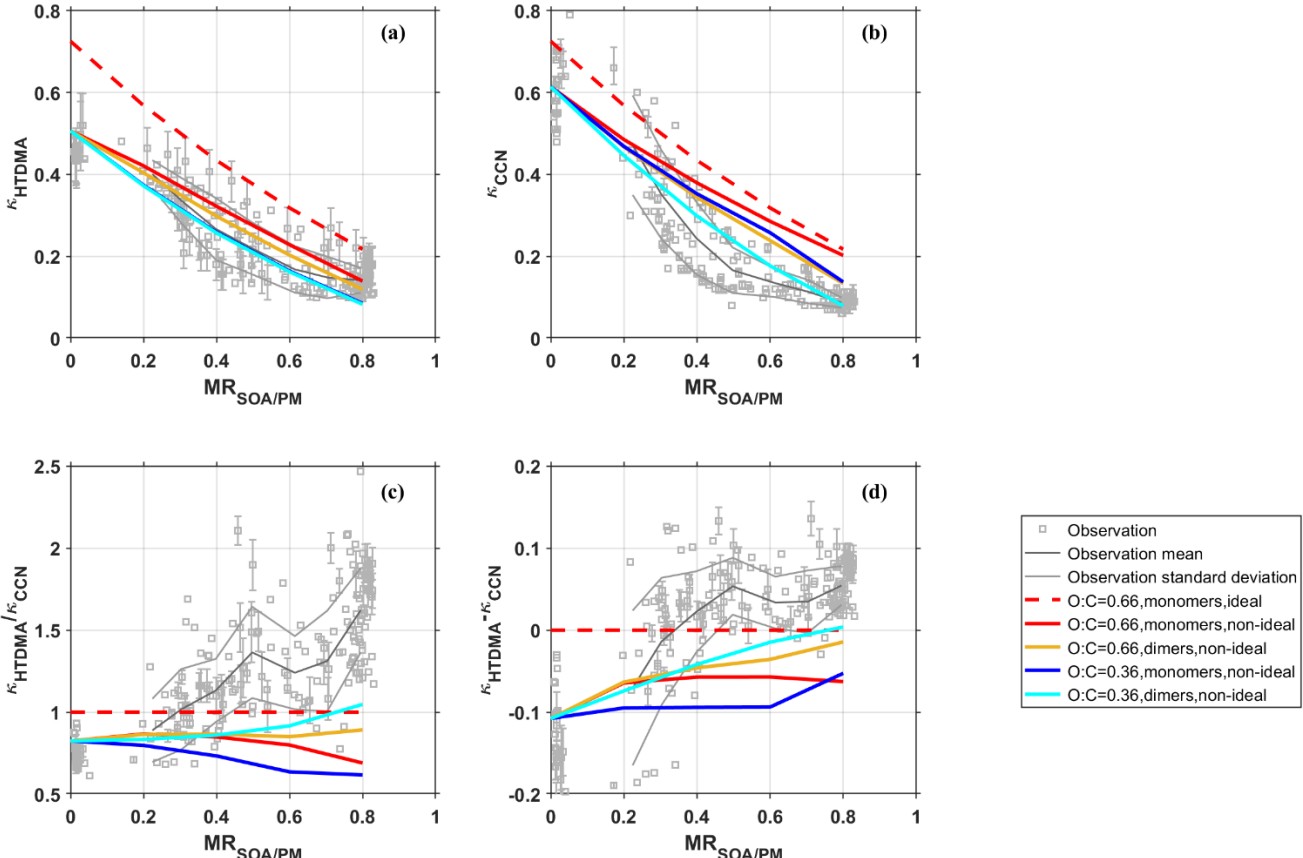

Figure 6: Influence of non-ideality on (a) $\kappa_{HTDMA}$, (b) $\kappa_{CCN}$, (c) $\kappa_{HTDMA}$ / $\kappa_{CCN}$, and (d) $\kappa_{HTDMA}$ - $\kappa_{CCN}$ analysed by comparison of model and experiment: Solid coloured lines show model results using AIOMFAC activity coefficients. The dashed red line shows the model result assuming an ideal solution for the same model compounds as the red solid line. The average O:C ratios of the model compound mixtures are given in the legend, the average molar masses are: 173 (red), 347 (yellow), 185 (blue) and 369 (cyan) g/mol. High molar masses were achieved by artificially dimerizing all organic compounds in the model calculations, labelled "dimers" in the legend. Grey dots and lines in the background show all experimental data points, and their mean and standard deviation, respectively. The uncertainty of the experimental data points is shown exemplarily for some points.