# Peer review of "On the evolution of sub- and super-saturated water uptake of secondary organic aerosol in chamber experiments from mixed precursors"

_Atmospheric Chemistry and Physics, 2021_

## Referee Comment (RC2)

This manuscript "On the evolution of sub- and super-saturated water uptake of secondary organic aerosol in chamber experiments from mixed precursors" presents a series of laboratory studies on SOA hygroscopicity at 90% RH and RH above 100 % under the different chamber conditions from single or multi-precursors. Comparison and discussion of hygroscopicity and CCN activity (k) of SOA are very important to understanding the aerosol-cloud interaction in atmosphere. Therefore, the manuscript fit well the scope of ACP. I recommendation its publication in ACP after addressing the following general and minor comments.

Minor comments:

1. Page 2: line 34-38 "With the increase of condensed SOA on seed particles throughout the experiments, the discrepancy of $\kappa_{HTDMA}$ and $\kappa_{CCN}$ became weaker (down to ~ 0 %) and finally the mean κHTDMA was ~ 60% higher than κCCN on average when the SOA mass fraction approached ~ 0.8. This is possibly attributable to the non-ideality of solutes at different RH or the different co-condensation of condensable organic vapours within the two instruments." The explanation is not clear. Also. I would suggest explain "non-ideality" of solutes at different RH.

2. Page 3: line 47-48 "The reliability of cloud condensation nuclei (CCN) activity predicted from the aerosol hygroscopic growth under sub-saturated condition remains unresolved" I would like to suggest that " the problem or issue of reliability….. remains unresolved"

3. Page 9: line 207-208 "Fig. 1 and Fig. 2" should be "Figure 1 and 2" please check other sentence in the whole of manuscript.

4. Page 12 line: 281-283 "As shown in Sec. 3.1-3.2, the aerosol chemical composition is size-dependent. It is essential to ensure the chemical composition are comparable for HTDMA and CCN measurements for the reconciliation study if their measured dry particle sizes are different" "are" should be "is"

5. Page 14 line 341-343: "It is worth noting that the prediction of critical diameter and CCN number concentration from $\kappa_{HTDMA}$ are based on the concurrently measured critical supersaturation and particle number size distribution." I would appreciate more explanation here.

6. Page 14 line 343-346: "The broader influences of the observed trend of $\kappa_{HTDMA}/\kappa_{CCN}$ as a function of MRSOA/PM on CCN activity prediction can vary a lot under different conditions of supersaturation and particle size distribution, which need further investigations."
What does author mean "under different conditions" please explain it.

General comments:

1. The authors should emphasize the novelty of the paper. The systems they chose have been somewhat studied by others in laboratory experiments. Comparison with the literature results are encouraged when appropriate.

2. What is the residence time for HTDMA system? As author discussed "non-ideality", does mean residence time affect SOA hygroscopicity, especially for formed viscous organic compounds?

Chan, M. N. and Chan, C. K.: Mass transfer effects in hygroscopic measurements of aerosol particles, Atmos. Chem. Phys., 5, 2703–2712, https://doi.org/10.5194/acp-5-2703-2005, 2005.

3. Mixing state effect on hygroscopicity and CCN activity of SOA under the sub-saturated and super-saturated condition are encouraged discussed in the main text. Have author perform experiments on morphology or mixing state of SOA at different RHs (e.g., RH<100% or RH>100%) using an optical microscopy or AFM?

4. Fig. 3 shows that there is a weak size dependence of GF (($\kappa_{HTDMA}$) between 100 and 200 nm aerosol particles, with up to ~ 0.2 due to non-uniform size-dependent chemical composition. Could author estimate uncertainty of GF (($\kappa_{HTDMA}$) at different sizes in the HTDMA system? or what is GF uncertainty of SOA aerosol nanoparticles with diameters from 75 to 300 nm at 90% RH as well as calculated $k_{HTDMA}$ uncertainty? Please see Table 2 from Mochida and Kawamura (2004).

5. Page 13 line: 302-304 "A higher $\kappa_{HTDMA}$ ($\kappa_{CCN}$) of the multi-component SOA-inorganic mixtures at the same level of MRSOA/PM indicated a higher $\kappa$ of the SOA, according to the ZSR mixing rule of $\kappa$ demonstrated in Petters and Kreidenweis (2007)." It is not clear, could author explain more in details? And could author evaluate $ZSR_{multi\text{-}component\ SOA\text{-}inorganic}$ according to Eq. (7) from Petter and Kreidenweis (2007)?

6. Page 13 line: 319-324: I would appreciate more discussion that how these factors (e.g., surface tension, molecular volume) affect the kappa, the following references are recommended:

Wang, J., Shilling, J. E., Liu, J., Zelenyuk, A., Bell, D. M., Petters, M. D., Thalman, R., Mei, F., Zaveri, R. A., and Zheng, G.: Cloud droplet activation of secondary organic aerosol is mainly controlled by molecular weight, not water solubility, Atmos. Chem. Phys., 19, 941–954, https://doi.org/10.5194/acp-19-941-2019, 2019.

Davies, J. F., Zuend, A., and Wilson, K. R.: Technical note: The role of evolving surface tension in the formation of cloud droplets, Atmos. Chem. Phys., 19, 2933–2946, https://doi.org/10.5194/acp-19-2933-2019, 2019.

---

## Author Comment (AC1)

**Response to comments of anonymous referees # 1, acp-2021-577**

**General comments**

In this paper, authors conducted a series of laboratory studies on the evolution of sub and super-saturated water uptake of secondary organic aerosol in chamber experiments from mixed precursors. The comparison of aerosol hygroscopicity under sub- and supersaturated conditions involves complex SOA characteristics are important for CCN predictions and understanding the RH effects (from sub-saturation to super saturation) on aerosol hygroscopicity and further improve aerosol hygroscopicity parameterizations. Therefore, the theme of this paper is scientifically meaningful, and the designed laboratory studies are also valid. However, more insightful analysis is required to better interpret the laboratory results.

*Many thanks to the reviewer for the comments and suggestions. We have improved the manuscript accordingly. Please find a point-by-point response below.*

Two main conclusions are drawn in this study.
The first one is that SOA composition played a second role in the kappa variations. I agree with the authors, and this is obvious considering that the well-known fact that inorganic aerosol is much more hygroscopic than SOA, thus, of course the SOA composition should play the second role. However, at the same MR_SOA/PM, considerable variations can be found, for example, at MR_SOA/PM ~0.4, the k_HTDMA varies from less than 0.2 to about 0.4, which demonstrates that the SOA compositions played significant roles in determining the apparent kappa. The authors should quantitatively derive the apparent/effective hygroscopicity parameter Kappa_OA using the volume mixing rule under sub-and supersaturated conditions and discuss their differences with previous results (Kuang et al., 2020b), quantitively present the relationships between Kappa_OA and SOA oxidation levels under sub- and super- saturated conditions and discuss more about the controlling factors of Kappa_OA variations based on AMS signals and VOC precursors (Wang et al., 2019). Otherwise, the authors are just talking about the sub- and super-saturated water uptake of mixed aerosol system includes organic and inorganic aerosols but not that of secondary organic aerosol, and is not consistent with the theme of the title.

*As suggested by both reviewers, I calculated the $\kappa_{org}$ with ZSR method, and included results/discussions including the suggested references into the manuscript. The paragraph has been rephrased as follows:*

*"Previous studies found the sub-saturated aerosol water uptake ($\kappa$) increases with chemical aging of SOA from single precursor oxidation and showed a positive relationship with SOA oxidation state (e.g. O:C ratio or $f_{44}$, fraction of m/z 44 in total organic signal) (Jimenez et al., 2009; Massoli et al., 2010; Lambe et al., 2011; Zhao et al., 2016; Duplissy et al., 2011; Kuang et al., 2020a), but no clear relationship involving multiple precursors with various oxidation state (Alfarra et al., 2013; Zhao et al., 2016). In addition, Wang et al. (2019a) found that the positive relation between water uptake at the super-saturated conditions and oxidation state (O:C) is attributed to lower molecular weight of organic species other than higher solubility at higher oxidation level. To illustrate relation between $\kappa$ of SOA and the oxidation state, the $\kappa_{org}$ was deduced with ZSR method and the $\kappa$ of ammonium sulphate from AIOMFAC assuming volume additivity. Two main messages are shown in Fig. S2. Firstly, the calculated $\kappa_{org}$ from HTDMA and CCN counter varied with VOC systems ranging from -0.2 to 0.2. The ZSR method assumes that components are independent and the water uptake by individual components are additive. Therefore, the negative values of the $\kappa_{org}$ indicates the existence of interactions between inorganic and organic substances and thus results in less water uptake than the case without interactions in ZSR method (Zardini et al., 2008). Secondly, the calculated $\kappa_{org}$ at sub- and super-saturated conditions showed no clear relationship with oxidation state of SOA ($f_{44}$) in various VOC systems, which is consistent with previous studies involving multiple precursors (Alfarra et al., 2013; Zhao et al., 2016). Other factors might have influences but need further investigations, such as organic mass loading, molecular weight (Cappa et al., 2011; Petters et al., 2017), solubility (Petters et al., 2009; Ruehl and Wilson, 2014; Huff Hartz et al., 2006), surface tension (Ovadnevaite et al., 2017; Bzdek et al., 2020; Ruehl et al., 2016; Lowe et al., 2019) and co-condensation (Kulmala et al., 1993; Topping et al., 2013b; Hu et al., 2018) and will be discussed in Section 3.5."*

[Figure]

Figure S2. (a) relation between $\kappa_{org}$ calculated from HTDMA and the fraction of m/z 44 in total organic signal, $f_{44}$. (b) relation between $\kappa_{org}$ calculated from CCN counter and $f_{44}$ in all investigated VOC systems.

The second one is "K_HTDMA/k_CCN increased as a function of SOA mass fraction, independent of initial VOC concentrations and sources, the mean k_HTDMA can be 60% higher than k_CCN on average when aerosol fraction approached 0.8". This finding is quite interesting but the explanation is not very convincing. The authors conclude that this finding is possibly attributable to the non-ideality of solutes at different RH or different cocondensation of condensable organic vapors within the two instruments. At L397, the authors claim that water increase in CCN set-up is not favorable for co-condensation of semi-volatile vapors. However, the condensation of these vapors such as HNO3 is influenced by both water vapor content and temperature. Although the temperature in CCN is higher than the temperature in HTDMA which seems not favorable for cocondensation, however, the aerosol water content in the CCN is much higher than in the HTDMA due to the super-saturated conditions. Thus, I cannot agree upon this argument. As to the possible role of no-ideality, the authors cited two papers published by Brechtel and Kreidenweis (2000a) and Brechtel and Kreidenweis (2000b) and said "the interactions of inorganic ions and organic molecules can exert both positive and negative effects in the water uptake, depending on the organic fraction

and inorganic species". The organic fraction and inorganic species are clear in this study, can authors perform some quantitative analysis? or at least deliver a clear qualitative result that the interactions of inorganic ions and organic molecules exerted a positive or a negative effect on the water uptake.

*Thanks for the reviewer. We agree that the evidence is ambiguous, and the discussion is not adequate. To better understand the role of non-ideality (and potentially co-condensation) in κ discrepancy between HTDMA and CCN, we performed AIOMFAC thermodynamic model calculations. Based on comparison between model results and the observation, a new section 3.5 has been added to understand how non-ideality and co-condensation influence sub- and super-saturated water uptake. Please find details in Section 3.5.*

*Towards the condensation of semi-volatiles (including HNO3), we agree with the reviewer that both temperature and water content influence this process. The competition of increasing temperature and dilution in aerosol phase due to increasing water content will influence the condensation/evaporation of the semi-volatiles in CCN counter. This discussion has been included into the updated manuscript.*

*The influence of non-ideality has been discussed in the updated manuscript. But the thermodynamic model cannot give information on organic/inorganic interactions as the reviewer suggested, which needs further explicit molecular information and worthwhile deeper investigations in the future.*

**Specific comments:**

1. L45-L50, just a suggestion: the first two sentences of the introduction lack continuity in logic, the authors jumped from aerosol-cloud interaction to the reliability of CCN prediction, I suggest a sentence that claim the accurate CCN prediction is essential for investigating aerosol-cloud interactions in climate models might be needed between the original two sentences.

*A sentence has been added as suggested: "Thus, an accurate prediction of cloud condensation nuclei (CCN) number from aerosol properties is essential for investigating aerosol-cloud interactions in climate models."*

2. L53, the SOA con be formed not just through gas-phase partitioning, but also aqueousphase reactions (Ervens et al., 2011;Kuang et al., 2020a).

*The sentence has been updated as suggested:*

*"A large portion of organic aerosols are secondary organic aerosol (SOA) (Zhang et al., 2007; Jimenez et al., 2009), formed from oxidation of gaseous volatile organic compounds (VOCs) via gas-particle partitioning (Hallquist et al., 2009) and aqueous-phase reactions (Ervens et al., 2011; Kuang et al., 2020)."*

3. L58, about the role of organic aerosol hygroscopicity in climate and aerosol cloudinteractions, references such as (Liu and Wang, 2010;Rastak et al., 2017) might be better choices.

*The sentence has been revised as suggested:*

*"Although the organic aerosol components are less soluble and consequently less hygroscopic than the referenced inorganic compounds (e.g. sulphate, nitrate) (Alfarra et al., 2013; Mcfiggans et al., 2006; Kreidenweis and Asa-Awuku, 2014; Huff Hartz et al., 2005; King et al., 2009), they can play an important role in the cloud formation globally (Liu and Wang, 2010; Rastak et al., 2017) due to its ubiquitous large fraction*

*(20 ~ 90 %) in fine particulate matter mass (Kanakidou et al., 2005; Jimenez et al., 2009; Zhang et al., 2007)."*

4. L164 more details about the DMA-CCN set-p should be given, for example, the detailed supersaturation points and time schedule

*Thanks for your suggestion.*

*More information about the setup has been added:*

*"During the experiments, DMA scans from 20 to 550 nm with 20 size bins. The selected aerosol particles will separate and go through the CCN counter and a CPC to measure the CCN and total particle number concentrations, respectively. The supersaturation ratio of CCN counter usually set to 0.5 % at the beginning of experiments. With the ongoing of SOA formation, the aerosol particles grow up. To derive a reliable activation curve with enough particle number concentration around the activation size, the set supersaturation ratio decreases accordingly down to 0.1 % during experiments, depending how fast the SOA is formed. The time resolution for each measurement is 10 mins."*

5. L175 Please report how the AMS vacuum aerodynamic diameter is converted to the mobility diameter, and estimate potential MR_SOA/PM uncertainties associated with this respect due to that obvious size-dependent chemical composition is observed. L226, a smaller particle size is not specific (compare to which size range), please presents diameter range directly

*For the conversion of AMS vacuum aerodynamic diameter to mobility diameter. Firstly, I estimate the density of the nonrefractory aerosol particles using simple mixing rule shown in equation [1] assuming the density of ammonium sulphate (1.77 g/cm$^3$) and SOA (1.4 g/cm$^3$ as used). $F_{m,SOA}$ is the mass fraction of the SOA. Then, this estimated density is used to calculate the mobility diameter as shown in equation [2] (Zhang et*

al., 2005).

$$\rho_{est} = \rho_{AS}(1 - F_{m,SOA}) + \rho_{SOA}F_{m,SOA} \quad [1]$$

$$D_m \approx \frac{D_{va}}{\rho_{est}} \quad [2]$$

For the potential $MR_{SOA/PM}$ uncertainty, the selection of SOA density can introduce uncertainty to the $\rho_{est}$, further on mobility diameter. Previous studies found that the SOA density can be different, with a range from 1.2 to 1.65 g/cm³ (Kostenidou et al., 2007; Alfarra et al., 2006; Varutbangkul et al., 2006; Nakao et al., 2013). For examples, Kostenidou et al. (2007) reported that the estimated density of SOA from α-pinene, β-pinene, d-limonene are 1.4-1.65 g/cm³. Nakao et al. (2013) investigated the SOA from 22 different precursors with a wide range of carbon number (C5-C15) and found their density ranges from 1.22 to 1.43, negatively related to their molecular size. In this study, considering the three precursors we used, we take a medium value of density (1.4 g/cm³) in the literature. To calculate the uncertainty of the SOA density on $MR_{SOA/PM}$, I recalculate with the minimum (maximum) density, 1.2 (1.65) g/cm³, the MR_SOA/PM changes within +/- 10%. The discrepancy of $MR_{SOA/PM}$ between 100 nm and 200 nm is outside of the uncertainty, which indicates the difference of chemical composition between these two sizes.

A new paragraph describing the diameter conversion and the uncertainty of density on MR_SOA has been added to the method sec. 2.2:

"For the conversion of AMS vacuum aerodynamic diameter to mobility diameter. Firstly, I estimate the density of the nonrefractory aerosol particles using simple mixing rule shown in equation [1] assuming the density of ammonium sulphate (1.77 g/cm³) and SOA (1.4 g/cm³ as used).

$$\rho_{est} = \rho_{AS}(1 - F_{m,SOA}) + \rho_{SOA}F_{m,SOA} \quad [1]$$

$F_{m,SOA}$ is the mass fraction of the SOA. Then, this estimated density is used to calculate the mobility diameter as shown in equation [2] (Zhang et al., 2005).

$$D_m \approx \frac{D_{va}}{\rho_{est}} \quad [2]$$

*For the potential MR_SOA/PM uncertainty, the selection of SOA density can introduce uncertainty to the $\rho_{est}$, further on mobility diameter. Previous studies found that the SOA density can be different, with a range from 1.2 to 1.65 g/cm³ (Kostenidou et al., 2007; Alfarra et al., 2006; Varutbangkul et al., 2006; Nakao et al., 2013). For examples, Kostenidou et al. (2007) reported that the estimated density of SOA from α-pinene, β-pinene, d-limonene are 1.4-1.65 g/cm³. Nakao et al. (2013) investigated the SOA from 22 different precursors with a wide range of carbon number (C5-C15) and found their density ranges from 1.22 to 1.43, negatively related to their molecular size. In this study, considering the three precursors we used, we take a medium value of density (1.4 g/cm³) in the literature. To calculate the uncertainty of the SOA density on $MR_{SOA/PM}$, I recalculate with the minimum (maximum) density, 1.2 (1.65) g/cm³, the $MR_{SOA/PM}$ changes within +/- 10%."*

*The sentence has been rephrased to clarify the size.*

[revised manuscript text omitted]

---

## Author Comment (AC2)

**Response to comments of anonymous referees # 2, acp-2021-577**

**General comments**

This manuscript "On the evolution of sub- and super-saturated water uptake of secondary organic aerosol in chamber experiments from mixed precursors" presents a series of laboratory studies on SOA hygroscopicity at 90% RH and RH above 100 % under the different chamber conditions from single or multi-precursors. Comparison and discussion of hygroscopicity and CCN activity (k) of SOA are very important to understanding the aerosol-cloud interaction in atmosphere. Therefore, the manuscript fit well the scope of ACP. I recommendation its publication in ACP after addressing the following general and minor comments.

*Many thanks to the reviewer for the comments and suggestions. We have improved the manuscript accordingly. Please find a point-by-point response below.*

**Minor comments:**

1.Page 2: line 34-38 "With the increase of condensed SOA on seed particles throughout the experiments, the discrepancy of κHTDMA and κCCN became weaker (down to ~ 0 %) and finally the mean κHTDMA was ~ 60% higher than κCCN on average when the SOA mass fraction approached ~ 0.8. This is possibly attributable to the non-ideality of solutes at different RH or the different co-condensation of condensable organic vapours within the two instruments." The explanation is not clear. Also. I would suggest explain "non-ideality" of solutes at different RH.

*Thanks for your suggestion. The reviewer 1 came up with the same question. To better understand the role of non-ideality for the κ discrepancy between HTDMA and CCN, we performed thermodynamic model calculations (AIOMFAC) and simulated the particle hygroscopic growth with inorganic/organic mixtures under ideal and non-ideal conditions. A new section 3.5 has been added to the updated manuscript to compare the simulation results with measurements.*

2. Page 3: line 47-48 "The reliability of cloud condensation nuclei (CCN) activity predicted from the aerosol hygroscopic growth under sub-saturated condition remains unresolved" I would like to suggest that " the problem or issue of reliability….. remains unresolved"

*The manuscript has been updated as suggested.*

3. Page 9: line 207-208 "Fig. 1 and Fig. 2" should be "Figure 1 and 2" please check other sentence in the whole of manuscript.

*This typo has been checked over the whole manuscript and corrected in the updated manuscript.*

4. Page 12 line: 281-283 "As shown in Sec. 3.1-3.2, the aerosol chemical composition is size-dependent. It is essential to ensure the chemical composition are comparable for HTDMA and CCN measurements for the reconciliation study if their measured dry particle sizes are different" "are" should be "is"

*This typo has been corrected in the updated manuscript.*

5. Page 14 line 341-343: "It is worth noting that the prediction of critical diameter and CCN number concentration from $\kappa_{HTDMA}$ are based on the concurrently measured critical supersaturation and particle number size distribution." I would appreciate more explanation here.

*We explained this in the next sentence (also your next comment), but as it did not seem to be sufficiently clear to understand. we have rephrased it for clarification:*

*"It is worth noting that the prediction of critical diameter and CCN number concentration from $\kappa_{HTDMA}$ are based on the concurrently measured critical supersaturation and particle number size distribution. This dependence of $\kappa_{HTDMA}/\kappa_{CCN}$ ratio on chemical composition can have a varied impact on the uncertainty of the*

*predicted CCN activity from sub-saturated $\kappa_{HTDMA}$ at different supersaturation ratio of water vapour and/or different particle number size distribution as measured above. Because the activated CCN number concentration is determined by all the three factors: the $\kappa_{HTDMA}$, water supersaturation ratio and particle size distribution. If at different supersaturation ratio of water vapour and/or different particle number size distribution as measured in this study, the uncertainty of the predicted CCN activity from sub-saturated $\kappa_{HTDMA}$ can change. Indeed, this discrepancy trend between $\kappa_{HTDMA}$ and $\kappa_{CCN}$ could introduce a varied impact on the CCN prediction, which needs further investigations."*

6. Page 14 line 343-346: "The broader influences of the observed trend of $\kappa$HTDMA/$\kappa$CCN as a function of MRSOA/PM on CCN activity prediction can vary a lot under different conditions of supersaturation and particle size distribution, which need further investigations."

What does author mean "under different conditions" please explain it.

*Here, "under different conditions of supersaturation" means different supersaturation ratio of water vapour. This sentence has been rephrased in the updated manuscript as shown in the 5th response above.*

General comments:

1. The authors should emphasize the novelty of the paper. The systems they chose have been somewhat studied by others in laboratory experiments. Comparison with the literature results are encouraged when appropriate.

*Sentences have been added in the introduction to emphasize the novelty of the paper:*

*"The novelty of the project is its design to investigate SOA formation from single to mixed precursors whereas previous studies mainly focused a single precursor (Voliotis et al., 2022). The interaction of the mixed precursor could influence SOA properties, therefore this study takes a further step of lab studies towards the real atmosphere where thousands of precursors are existing and reacting at the same time even the chemical regime and complexity of the chamber studies could deviate from the real atmosphere.*

*In the updated manuscript, a new section on AIOMFAC model simulations and the $\kappa$ of SOA at sub- and super-saturated conditions based on ZSR method are added. More comparisons with literature and discussions are added to explain the observation. Please see details in section 3.5.*

2. What is the residence time for HTDMA system? As author discussed "non-ideality", does mean residence time affect SOA hygroscopicity, especially for formed viscous organic compounds?

Chan, M. N. and Chan, C. K.: Mass transfer effects in hygroscopic measurements of aerosol particles, Atmos. Chem. Phys., 5, 2703–2712, https://doi.org/10.5194/acp-5-2703-2005, 2005.

Explain the non-ideality.

*As demonstrated in Chan and Chan (2005), a few seconds of residence time in HTDMA system (30s in this study, (Duplissy et al., 2009)) is possibly not enough for aerosol particles with viscous organic film to equilibrate with water vapour. The inhibited water*

*uptake then leads to underestimation of aerosol hygroscopic growth. In my recent work, we concurrently recorded the particle rebounding fraction as an indicator of phase state and found that aerosol particles tend to be liquid-like without rebounding when RH is larger than 80% irrespective of the organic mass fraction (0-0.8) in all investigated VOC systems (Wang et al., 2021). Therefore, at 90% RH, the equilibration of water should be fast. Furthermore, the residence time can influence gas-particle partitioning of semi-volatile species. We address the topic of semi-volatility in Sec. 3.5 of the revised manuscript.*

*The detailed discussion on how non-ideality influences the hygroscopic growth is also shown in the newly added section 3.5.*

3. Mixing state effect on hygroscopicity and CCN activity of SOA under the sub-saturated and super-saturated condition are encouraged discussed in the main text. Have author perform experiments on morphology or mixing state of SOA at different RHs (e.g., RH<100% or RH>100%) using an optical microscopy or AFM?

*We fully agree that the morphology or mixing state measurement will largely improve our discussion, but unfortunately it is not available in this study. It is a very interesting topic, which is worthwhile to investigate in the future study.*

4. Fig. 3 shows that there is a weak size dependence of GF (($\kappa$HTDMA) between 100 and 200 nm aerosol particles, with up to $\sim$ 0.2 due to non-uniform size-dependent chemical composition. Could author estimate uncertainty of GF (($\kappa$HTDMA) at different sizes in the HTDMA system? or what is GF uncertainty of SOA aerosol nanoparticles with diameters from 75 to 300 nm at 90% RH as well as calculated kHTDMA uncertainty? Please see Table 2 from Mochida and Kawamura (2004).

*Good suggestions. An uncertainty estimation is important when comparing GF/$\kappa$ values*

*among different particle sizes, experiments and instruments. The uncertainty of κ values from HTDMA and CCN counter are estimated following the method in Irwin et al. (2010). Herein, for HTDMA measurement, the GF uncertainty is calculated with the same method as in Mochida and Kawamura (2004) as the reviewer suggested and then propagated to the calculation of κ. Overall, the κ uncertainty for HTDMA and CCN are within 10%. The κ uncertainty has been added to Fig.5 and the discussion in Sec. 3.5.*

5. Page 13 line: 302-304 "A higher $\kappa_{HTDMA}$ ($\kappa_{CCN}$) of the multi-component SOA-inorganic mixtures at the same level of $M_{RSOA}/PM$ indicated a higher κ of the SOA, according to the ZSR mixing rule of κ demonstrated in Petters and Kreidenweis (2007)." It is not clear, could author explain more in details? And could author evaluate ZSRmulti-component SOA-inorganic according to Eq. (7) from Petter and Kreidenweis (2007)?

*Thanks for your comment. The reviewer 1 also pointed out this (1ˢᵗ question in the general comment). I have calculated the κ of OA with ZSR method as both of you suggested and discussed in the updated manuscript. Please see the added part in Sec. 3.4.*

6. Page 13 line: 319-324: I would appreciate more discussion that how these factors (e.g., surface tension, molecular volume) affect the kappa, the following references are recommended:

Wang, J., Shilling, J. E., Liu, J., Zelenyuk, A., Bell, D. M., Petters, M. D., Thalman, R., Mei, F., Zaveri, R. A., and Zheng, G.: Cloud droplet activation of secondary organic aerosol is mainly controlled by molecular weight, not water solubility, Atmos. Chem. Phys., 19, 941–954, https://doi.org/10.5194/acp-19-941-2019, 2019.

Davies, J. F., Zuend, A., and Wilson, K. R.: Technical note: The role of evolving surface tension in the formation of cloud droplets, Atmos. Chem. Phys., 19, 2933–2946, https://doi.org/10.5194/acp-19-2933-2019, 2019.

*All impacting factors are discussed in the new added section 3.5, including e.g. surface tension, molecular weight. Your suggested references provide useful evidence and have been cited in the discussions.*

References of response:

Chan, M. N. and Chan, C. K.: Mass transfer effects in hygroscopic measurements of aerosol particles, Atmos. Chem. Phys., 5, 2703-2712, 10.5194/acp-5-2703-2005, 2005.

Duplissy, J., Gysel, M., Sjogren, S., Meyer, N., Good, N., Kammermann, L., Michaud, V., Weigel, R., Martins dos Santos, S., Gruening, C., Villani, P., Laj, P., Sellegri, K., Metzger, A., McFiggans, G. B., Wehrle, G., Richter, R., Dommen, J., Ristovski, Z., Baltensperger, U., and Weingartner, E.: Intercomparison study of six HTDMAs: results and recommendations, Atmos. Meas. Tech., 2, 363-378, 10.5194/amt-2-363-2009, 2009.

Irwin, M., Good, N., Crosier, J., Choularton, T. W., and McFiggans, G.: Reconciliation of measurements of hygroscopic growth and critical supersaturation of aerosol particles in central Germany, Atmos. Chem. Phys., 10, 11737-11752, 10.5194/acp-10-11737-2010, 2010.

Seinfeld, J. H. and Pandis, S. N.: Atmospheric chemistry and physics : from air pollution to climate change, Third edition., John Wiley & Sons, Hoboken, New Jersey2016.

Wang, Y., Voliotis, A., Shao, Y., Zong, T., Meng, X., Du, M., Hu, D., Chen, Y., Wu, Z., Alfarra, M. R., and McFiggans, G.: Phase state of secondary organic aerosol in chamber photo-oxidation of mixed precursors, Atmos. Chem. Phys., 21, 11303-11316, 10.5194/acp-21-11303-2021, 2021.